# Numerical simulation of a rapidly developing bow echo over northeastern Poland on 21 August 2007 using near-grid-scale stochastic convection initiation

Damian K. Wójcik[1,2], Michał Z. Ziemiański[2], Wojciech W. Grabowski[3]

[1]Department of Atmospheric and Cryospheric Sciences, University of Innsbruck, Innsbruck, Austria
[2]Institute of Meteorology and Water Management - National Research Institute, Warsaw, Poland
[3]NSF National Center for Atmospheric Research, Boulder, Colorado

*Correspondence to*: Damian K. Wójcik, damian.wojcik@uibk.ac.at, damian.wojcik@imgw.pl

**Abstract.** A rapidly developing fast propagating meso-β-scale severe bow echo that developed over northeastern Poland on 21 August 2007 caused significant property damage and resulted in 12 fatalities. The operational model of Consortium for Small Scale Modeling (COSMO) with a horizontal grid spacing of 2.2 km is used for its numerical simulation but encounters significant problems despite favorable environmental conditions. The implementation of a new stochastic convection initiation scheme in a 9-member ensemble enables the numerical reconstruction of the event as the cold-pool-driven convective system, with peak gusts closely matching the observed values. The scheme uses small-scale temperature perturbations arguably resembling near-grid-scale convective cells developing within the boundary layer. The perturbations influence not only CAPE and CIN, but also the lower-tropospheric vertical wind shear. Initial and boundary conditions for the experiment are based on ERA5 reanalysis. Additional data assimilation of local surface observations improves the reconstruction of atmospheric environmental conditions. A supplementary experiment tests the forecast sensitivity to an increase of low-to-mid tropospheric winds, and thus the vertical shear, and shows an increase of the maximum surface gusts within the ensemble when the convection initiation is implemented. The simulations' main drawbacks are about an hour delay with respect to observations in the development of maximum gusts and a tendency to produce isolated convective cells along the leading edge of the system's cold pool rather than a more coherent structure observed within the bow echo.

## 1 Introduction

Bow echoes (Fujita, 1978; Przybylinski, 1995) are a specific class of deep moist convection self-organization. They often produce high-impact weather, especially in the form of damaging winds and sometimes tornadoes (Fujita, 1978; Johns and Hirt, 1987; Gallus et al., 2008). However, bow echoes are especially difficult for numerical representation and prediction (Snively and Gallus, 2014; Lawson and Gallus, 2016). Bow echoes are observed relatively frequently over the US (Klimowski et al., 2003) and Europe (Pacey et al., 2021), including Poland (Celiński-Mysław and Palarz, 2017). As discussed in Klimowski et al. (2004), bow echoes may originate from a weakly organized group of cells, squall lines, or supercells. Their horizontal extent varies between 10 to 25 km for a single-cell bow to hundreds of kilometers for organized systems. Their lifespan varies from about 1 hour for the former to many hours for the latter (Klimowski et al., 2004). A special class of bow echoes is associated with derecho events that feature the length of the damage swath of downburst clusters of several hundred kilometers (Johns and Hirt, 1987; Corfidi et al., 2016).

The idealized modeling studies indicate that bowing systems tend to form on the leading edge of convective cold pools in the presence of significant vertical wind shear (Weisman and Klemp, 1986; Weisman, 1992; 1993). A

theory by Rotunno et al. (1988) (known as the RKW theory) explains the systems' organization and persistence via the approximate balance of horizontal vorticity of the environmental flow and of the sufficiently strong cold pool flow at the pool's leading edge. The balance forces deep lifting and a formation of new convective cells at the leading edge, making bow echoes cold-pool-driven (Coniglio et al., 2005) systems. Further research confirmed the validity of the RKW theory for idealized systems (see Bryan et al., 2006). The studies of real events also

confirm the presence of the RKW mechanism for strong low-level shears, while indicating that also the presence of a notable shear above may lead to the development of such severe and persistent systems (Stensrud et al., 2005; Weisman and Rotunno, 2005; Cognilio et al., 2012; Kirshbaum et al., 2025). Mature organized linear convection also tends to develop a mid-tropospheric inflow current, the rear inflow jet (RIJ), accelerated by the horizontal pressure gradient from condensational heating on the systems' leading edge (LeMone, 1983; Smull and Houze,

1987). Idealized modeling studies indicate an important role of RIJ for the bow echoes' persistence and the downward horizontal momentum transport (Weisman, 1992; 1993; Mahoney et al., 2009).

An ability of realistic numerical representation of such high-impact convective systems with contemporary convective-scale numerical weather prediction (NWP) models (Baldauf et al., 2011; Done et al., 2004; Saito, 2012; Yano et al., 2018) is important both for research and operational forecasting. Numerical models have already been

used for successful numerical case studies of bow echoes developing over the US (e.g., Weisman et al., 2013; Xu et al., 2015; Parker et al., 2020; Liu et al., 2023) and Europe (Toll et al., 2015; Mathias et al., 2017), including Poland (Taszarek et al., 2019; Figurski et al., 2021; Kolonko et al., 2023; Mazur and Duniec, 2023), Africa (Diongue et al., 2002), and Asia (Meng et al., 2012; Xu et al., 2024). However, such studies encounter several problems. Some stem from the inherent limited predictability of atmospheric processes at convective scales

(Lorenz, 1969; Vallis, 2006; Palmer et al., 2014; see also a discussion in Lawson and Gallus, 2016). On the mesoscale, the predictability may be increased by a strong external forcing, for instance, by the underlying surface or by large-scale disturbances (Anthes et al., 1985). It is thus no surprise that the successful bow echo numerical simulations concern mainly systems prone to increased predictability: relatively large or long-lasting (6 or more hours, including derechos), embedded within large convective systems, or developing under a significant external

forcing (e.g., by fronts, Lawson and Gallus, 2016).

Other significant limitations stem from the models' uncertainties in representing grid- and sub-grid-scale processes (see extended discussions in Lawson and Gallus, 2016; and Varble et al., 2020). Bryan et al. (2003) and Lebo and Morrison (2015) show that the representation of deep convective updrafts requires at most 0.25 km horizontal grid spacing. Convective currents are therefore highly under-resolved in contemporary NWP models that typically

feature horizontal grid spacing of O(1 km). The need to parameterize sub-grid processes introduces inevitable uncertainties as well. For instance, the representation of cloud microphysics significantly influences stratiform precipitation and convective updraft strength (e.g., Varble et al., 2014a; b). Also, the uncertainties related to the representation of turbulence affect cloud organization (Machado and Chaboureau, 2015; Tompkins and Semie, 2017). The overall impact of those model uncertainties on the simulated evolution of organized convective systems

is still unclear (Varble et al.; 2020).

The convective-scale simulation ensembles (e.g., Gebhardt et al., 2008; Clark et al., 2011; Hacker et al., 2011; Bouttier et al., 2012) aim to account for the forecast uncertainties. They typically apply perturbations to boundary and initial conditions (BC and IC, respectively) to account for large-scale uncertainties while the model

ambiguities are represented by methods like stochastically perturbed parameterization tendencies (Buizza et al., 1999) or the stochastic kinetic energy backscatter (SKEB, Shutts, 2005). Such ensembles are also used for bow echo studies (e.g., Melhauser and Zhang, 2012; Lawson and Gallus, 2016; Grunzke and Evans, 2017; Lawson et al., 2020; Ribeiro et al., 2022), although they typically focus on long-lived systems.

From the process-level viewpoint, one of the main problems of convective-scale simulations is an insufficient representation of convection initiation (CI), especially for weak external forcings (Kühnlein et al., 2014; Clark et al., 2016; Hirt et al., 2019). To alleviate the problem, Hirt et al. (2019) and Puh et al. (2023) tested physically-based stochastic perturbations (PSP, Kober and Craig, 2016) of the temperature, vertical velocity, and humidity. Zeng et al. (2020) used the PSP and perturbations in the form of warm bubbles to account for the model errors in data assimilation, while Clark et al. (2021) and Flack et al. (2021) used the PSP to represent boundary layer variability. These studies are mainly interested in statistically measured overall impact of the schemes on precipitation and some atmospheric-state parameters, but not in a reconstruction of a specific convective system development, we are interested in. The studies agree that the PSP has the largest positive impact on weakly-forced (non-equilibrium; see e.g. Emanuel, 1994; Done et al., 2006) convection. It generally increases ensemble spread, but its impact on prognostic parameters is described as limited: Puh et al. (2023) conclude that it "improves the spatial distribution of precipitation slightly" and for "near-surface variables predominantly shows a neutral to slightly beneficial forecast performance". As for the warm bubbles' technique (Zeng et al., 2020), it is not used as a stochastic method but to assimilate already existing radar-detected convective developments not recognized by the model forecast. It does not increase ensemble spread but improves precipitation forecasts up to 3 hours, like the PSP.

The above studies (except a side-experiment by Clark et al., 2021) apply perturbation horizontal sizes of O(10 km). This is the consequence of diffusive properties of NWP models, which significantly damp the flow modes having scales smaller than the model's effective resolution size (Skamarock, 2004), usually in the range of 6 to 8 grid lengths. Thus, a paradigm was accepted, also for the PSP methods (Kober et al., 2016; Clark et al., 2021), that the model perturbations should have horizontal sizes of at least the effective resolution size, and the vast experience of ensemble forecasting confirmed that practice (Palmer, 2019). Here, we want to experiment with a CI perturbation tactic that aims at stirring the flow variability at the near-grid scales, below the effective resolution size. That scale has a strong physical justification, being the scale of observed large boundary layer thermals with horizontal sizes of 1 to 3 km (William and Hacker, 1992, 1993; Marquis et al., 2021, and references therein; see also Grabowski, 2023) and initial convective cells with horizontal sizes of 3 to 5 km (Marquis et al., 2021). We demonstrate that such a scheme, used in a contemporary convective-scale NWP model, with likely overestimated perturbation amplitudes compensating for their unphysical damping, facilitates the numerical representation of a high-impact, rapidly developing, isolated bow echo of Orlanski's (1975) meso-β-scale as the cold-pool-driven convective system with maximum gusts close to the observed ones, as long as correct large-scale environmental conditions are used. We aim not only at a realistic representation of the convective event but are also interested in how the CI influences (model-represented) atmospheric processes responsible for deep convective development. For the latter, we analyse the CI's impact on the convective properties of the atmosphere (CAPE, CIN, low-level shear) and their further impact on the developing severe convection. These matters were not addressed in the previous studies.

The study's object is an isolated bow echo that developed within a warm air mass over northeastern Poland on 21 August 2007 (Fig. 1), caused substantial damages and had an exceptionally strong social impact with 12 fatalities

(compare with Surowiecki and Taszarek, 2020). Our process-oriented strategy reproduces the large-scale atmospheric environment in a deterministic sense, applying available reanalysis and observations, and uses a 9-member ensemble, driven only by randomly generated CI perturbations. The use of such a CI-oriented ensemble is new for bow echo studies but reminds Lawson et al. (2020) who used only SKEB perturbations. Standard COSMO (Consortium for Small Scale Modeling) NWP model (Baldauf et al., 2011) and ERA5 reanalyses

(Hersbach et al., 2020) are used.

An additional novelty is in augmenting the results of global ERA5 reanalysis with mesoscale data assimilation (DA), using available surface observations. That is inspired by operational procedures of augmenting global DA with regional DA for regional NWP (Gustafsson et al., 2018; Baldauf et al., 2011; Bučánek and Brožková, 2017; Müller et al., 2017). However, while regional DA mainly provides information on possibly small-scale flow

variability, it is not the case here with the ERA5 31-km horizontal grid size and a 70 km characteristic distance between weather stations over Poland. Instead, we aim at a correction of larger-scale systematic temperature errors over northeastern Poland. We use operational COSMO nudging (Schraff and Hess, 2021) for that purpose. The aerological soundings are not additionally assimilated mainly due to problems with their representativity for the environment of the developing convection (see Section 2.2).

After analysing the convective system's environmental conditions and development, we demonstrate that the standard convective-scale COSMO operational NWP model without the CI scheme has significant problems with forecasting the system even with realistic environmental conditions (in terms of CAPE, CIN, and increased low-level vertical wind shear). After implementing the stochastic CI, we analyze its influence on the environmental conditions, including CAPE, CIN, and shear. Next, we analyze the dynamics of the developing convection,

focusing on the cold-pool-driven mechanism and RIJ formation. An additional experiment investigates how a slight increase in low-to-mid tropospheric wind shear influences the dynamics of convective development. A comparison of the experiments' results with available observations allows to identify inevitable imperfections of the system's numerical representation.

The paper is organized as follows. Section 2 describes the pre-storm weather conditions and the development of

the bow echo system. Section 3 presents the data and models used for the study, and discusses a reconstruction of the initial conditions for the prognostic experiments. Section 4 introduces the CI scheme and presents its impact on the developing convection. Additional simulations involving increased low-to-mid tropospheric wind shear are discussed in Section 5. Section 6 discusses the development of the RIJ in the forecast most closely resembling the actual system development. The paper ends with the summary and conclusions in Section 7. Supplement contains

additional figures supporting the discussion of Sections 4 and 5 (for most of "not shown" remarks).

**2 The bow echo and its environmental conditions**

**2.1 Synoptic overview**

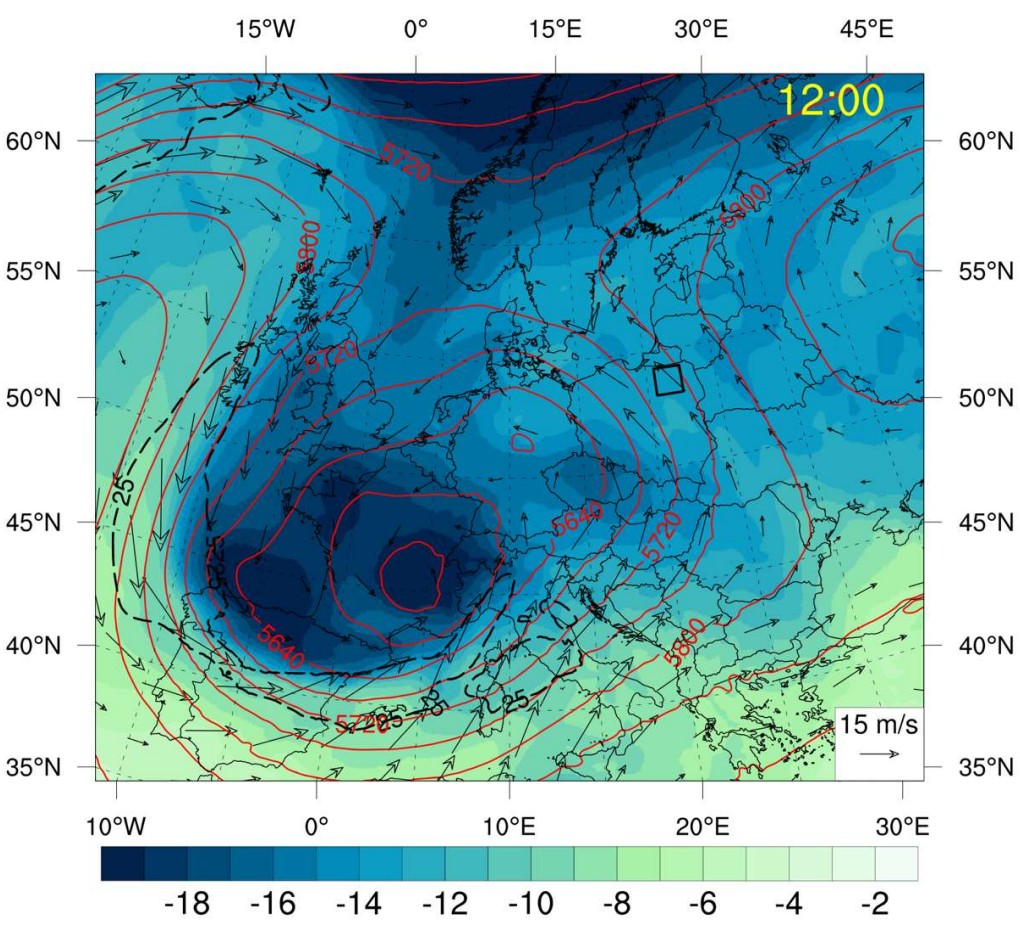

Figure 1: ERA5-based analysis at 500 hPa, on 21 August 2007, 12:00 UTC: temperature (°C in colors), altitude (in m in red contours), wind vectors and 25 m s⁻¹ isotach (black dashed contour); the black rectangle indicates the area of convective development over northeastern Poland.

On 21 August 2007, eastern Poland was covered by a warm subtropical Mediterranean air mass, away from the cold front situated near Poland's western border (not shown). The area was located between a deep cold upper-level low over western Europe and a warm ridge from over central European Russia (see Figs. 1 and 2 for ERA5-based analysis at 500 hPa; the data are processed using int2lm preprocessor of the COSMO Consortium, Schättler and Blahak, 2021). A synoptic-scale tropospheric thermal gradient over Poland induced relatively strong upper-level southerly to southeasterly winds. At 12:00 (all hours are given in UTC and are presented in the format HH:mm, where HH are hours and mm are minutes; 00:00 UTC corresponds with 02:00 Central European Summer Time), they reached 12-13 m s⁻¹ at 700 hPa (not shown) and 16 m s⁻¹ at 500 hPa over northeastern Poland (see Fig. 2b for a localized band of stronger wind in the area), but less than 14 m s⁻¹ at 300 hPa (the jet stream is located further southwest, not shown). There is a lack of notable warm advection at 700 (not shown) and 500 hPa (Fig. 2a). Between 12:00 and 15:00, there is a prevailing weak negative relative vorticity advection at 500 hPa over the area (Fig. 2c and 2d), despite a distant presence of a short-wave trough approaching from the south. That suggests a weak dynamic suppression of deep convective activity over the area (following the omega equation interpretation, e.g., Holton, 2004).

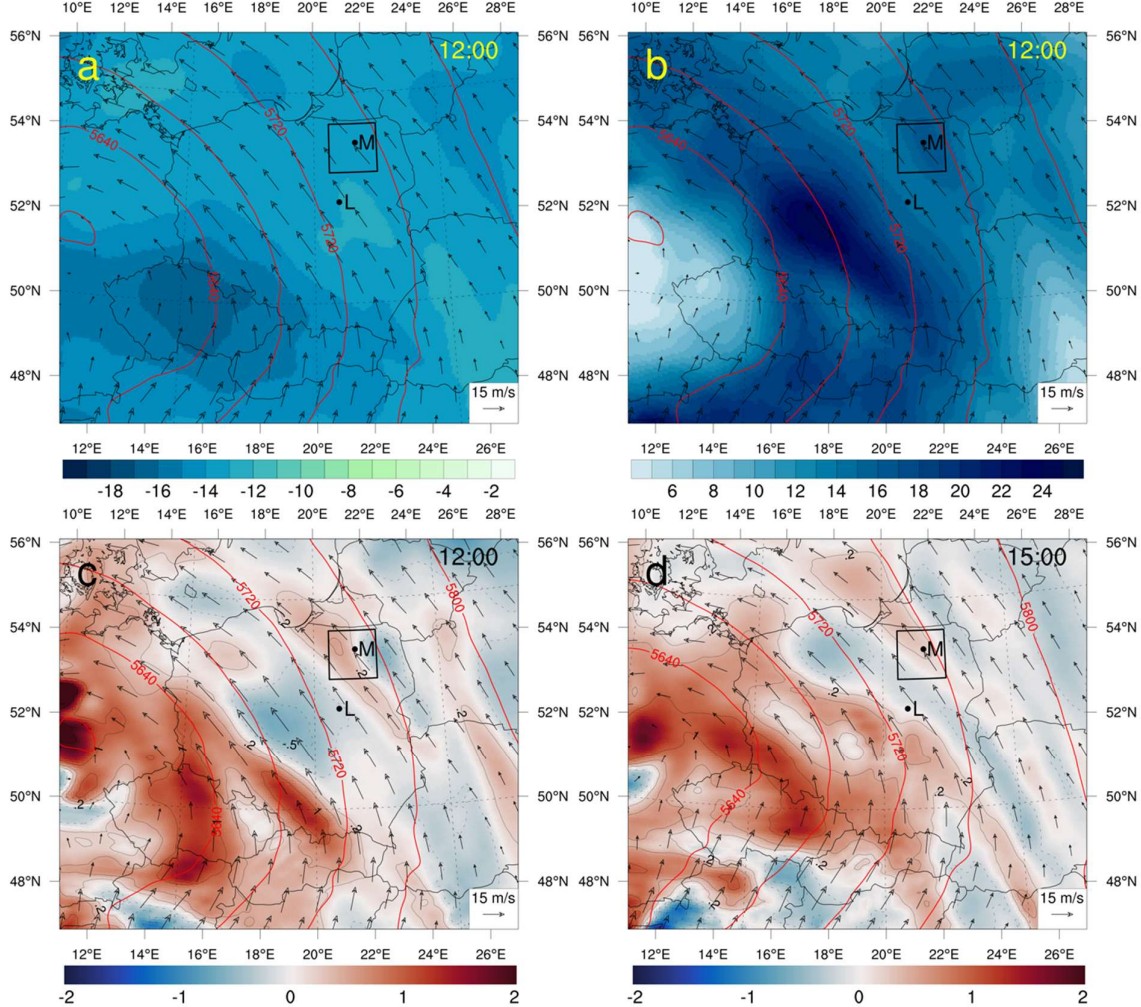

**Figure 2: ERA5-based analysis at 500 hPa, on 21 August 2007: (a) 12:00 UTC temperature (°C in colors), altitude (in m in red contours), and wind vectors; (b) 12:00 UTC wind speed (in m s⁻¹ in colors), altitude (red contours) and wind vectors; (c) 12:00 UTC relative vorticity (in units $10^{-4}$ s⁻¹ in color and contours at plus/minus 0.2, 0.5, 1.0, and 2.0 units, negative values are dashed), altitude (red contours) and wind vectors; (d) as (c) but at 15:00 UTC. The black rectangle indicates the area of convective development over northeastern Poland, and the black dots indicate the positions of weather stations at Legionowo (L) and Mikołajki (M).**

## 2.2 Mesoscale conditions preceding severe convection development

Between 00:00 and 09:00 on 21 August 2007, northeastern Poland remained free from convective activity (not shown). The local synoptic weather stations (WSs) at Mikołajki and Kętrzyn (locations shown in Fig. 3b) reported high morning 2-m dewpoint temperatures ($T_d$) in the range of 18 to 20°C. Further presence of very moist air in the area was documented by a surface analysis at 12:00 (Fig. 3b), showing a relatively narrow moist plume covering Mikołajki (maximum $T_d$ of 20.2°C) and elongated toward the north-northwest, towards Kętrzyn. With increasing daytime insolation, the formation of a typical convective boundary layer (CBL) was expected over the area.

The vertical air-mass structure (Fig. 3a) was probed by a radiosonde ascent launched at 11:16 at Legionowo, 120 km southwest of the convection initiation area. The sounding showed an 800-m-deep well-mixed, but relatively dry (2-m $T_d$ of 15.9°C) CBL capped by a layer with a reduced temperature lapse rate. The wind speed increased with height from 1 m s⁻¹ at about 100 m AMSL to almost 10 m s⁻¹ at 650 m AMSL with east to south-southeast directions. Above, the wind speed increased from 9.5 m s⁻¹ at 700 hPa to 18 m s⁻¹ at 500 hPa and 23 m s⁻¹ at

300 hPa, and wind directions were from east-southeast to southeast. Moderate wind-speed shear of around 9 m s$^{-1}$ was observed between the surface and 700 hPa level, slightly above the 25 percentile for climatology of warm-season bow echo systems over Poland (Celiński-Mysław et al., 2018).

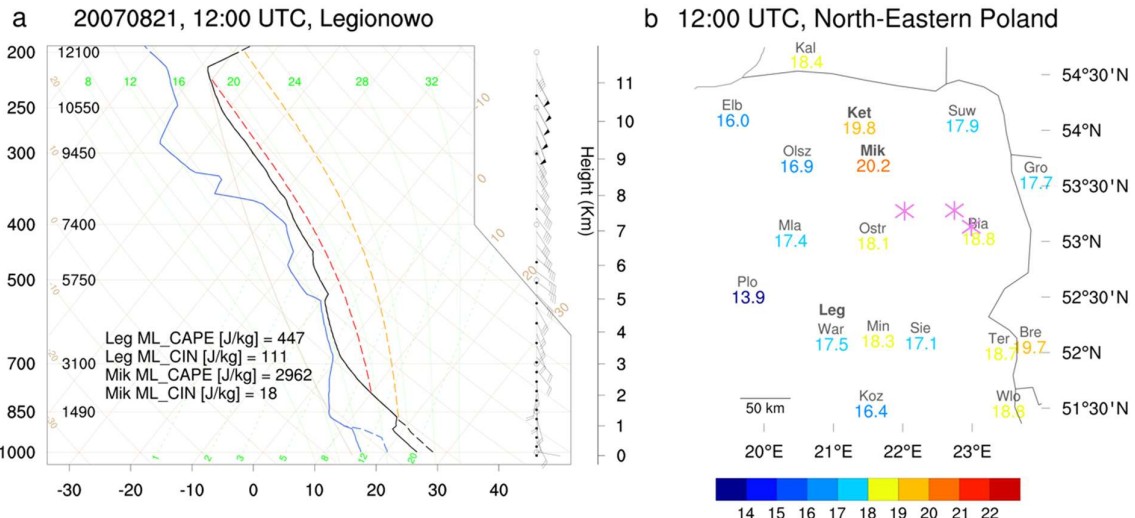

**Figure 3: Skew-T diagram and wind (a, left) from Legionowo ascent on 21 August 2007 at 12:00 and (b, right) 2-m dewpoint temperatures (color scale and numerical values in °C) over northeastern Poland at that time. In (a), the black and blue continuous lines show T and T$_d$ (°C), respectively, and the black and blue dashed lines show T and T$_d$ reconstructed for CBL over Mikołajki (see text), respectively. The dashed red and yellow lines show results of pseudo-adiabatic ascent for Legionowo and Mikołajki, respectively. The resulting values of MCAPE and MCIN are shown in the lower-left corner of the figure. In (b), the abbreviations denote weather stations: Kaliningrad (Kal), Elbląg (Elb), Kętrzyn (Ket), Suwałki (Suw), Olsztyn (Olsz), Legionowo (Leg), Mikołajki (Mik), Grodno (Gro), Mława (Mla), Ostrołęka (Ost), Białystok (Bia), Płock (Plo), Warszawa (War), Mińsk Mazowiecki (Min), Siedlce (Sie), Terespol (Ter), Brest (Bre), Kozienice (Koz), and Włodawa (Wlo). Three purple asterisks mark the locations of the secondary convection initiation (compare Fig. 3).**

The sounding-derived mixed layer parcel (with properties averaged over the lowest 500 m) convective inhibition (MCIN) was quite large at 111 J kg$^{-1}$, while the mixed layer parcel convective available potential energy (MCAPE) was relatively small at 447 J kg$^{-1}$, the latter significantly below the 25 percentile of MCAPE for warm-season bow echo systems over Poland of about 700 J kg$^{-1}$ (Celiński-Mysław et al., 2018). Further northeast, however, the conditions were more supportive due to the observed significantly higher 2-m T and T$_d$ (Fig. 3b). These conditions can be assessed by constructing a thermodynamic diagram linking well-mixed CBL characteristics estimated from local surface observations (following McGinley, 1986; and approximating CBL temperature and humidity profiles with dry adiabatic lapse rate and constant water vapor mixing ratio, respectively) and free tropospheric structure taken from the Legionowo sounding. Such a reconstructed sounding for Mikołajki (Fig. 3a, see also Wójcik, 2021) estimates MCIN and MCAPE at around 18 and 2900 J kg$^{-1}$, respectively, favoring strong convection.

## 2.3 Convection development

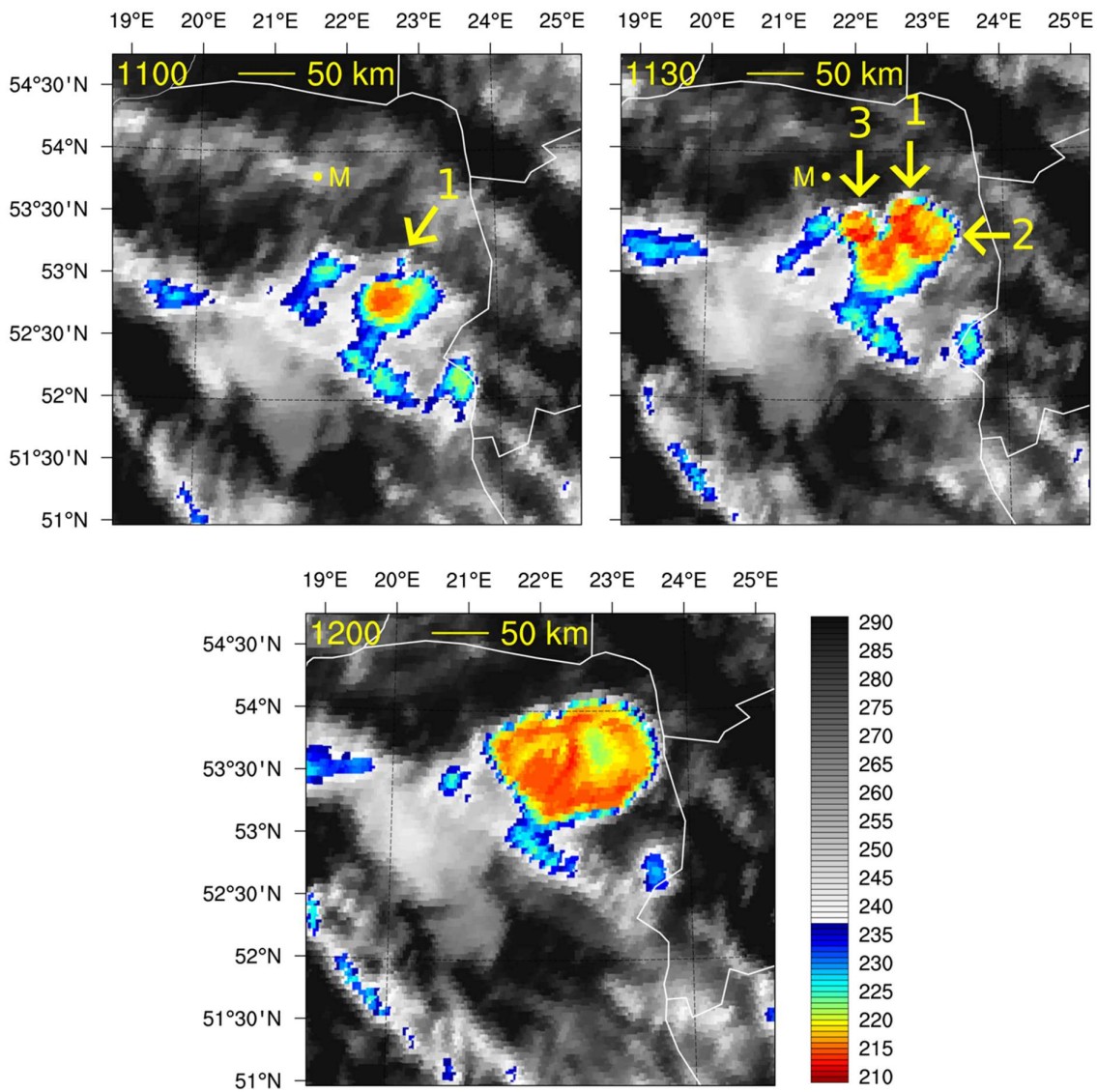

**Figure 4: Convection initiation over northeastern Poland using brightness temperature derived from 10.8 μm channel observations of the Meteosat satellite (EUMETSAT). Numbers 1 to 3 denote individual towering cumulus clouds developing around 11:00 with arrows pointing toward the clouds. The yellow dot denotes the location of Mikołajki (M), the time in UTC is in the upper-left corner of every plate, yellow line shows a distance of 50 km.**

The deep convection initiation over northeastern Poland on 21 August took place outside the available radar range and is documented using Meteosat 10.8 μm imagery (Fig. 4). At 10:45, a strong primary convective cell was present south-southeast of Mikołajki with a cloud top of a few tens of kilometers wide and brightness temperatures below 220 K (not shown). The secondary convection initiated shortly before 11:00, with a new towering convective cloud (marked as 1 in Fig. 4) seen immediately north of the primary cell. Subsequently, two convective cells developed nearby at 11:15, marked as 2 and 3 in Figure 4 at 11:30. The anvils of the developing cells merged, producing at 12:00 an extensive (~100 km across) cloud with the cloud-top brightness temperature below 220 K. The system moved towards northwest and, near 12:00, started to enter the range of the meteorological radar from Gdańsk (located west-northwest of the system). The radar indicated (Fig. 5) that the system's shape started to resemble the C-stage bow echo of Fujita (1978) around 13:00. That stage was also observed at 14:00 and 15:00.

The radar-derived system propagation speed reached approximately 23 m s$^{-1}$ between 13:00 and 14:20. The propagation direction (approx. 320°) agreed with wind direction in the Legionowo sounding at 700 and 500 hPa (Fig. 3) and was close to the orientation of the surface layer moisture plume, maintaining a moisture supply for the system development. The length of the bowing segment reached around 85 km at 13:30 and around 120 km at 235 15:00 (Fig. 5).

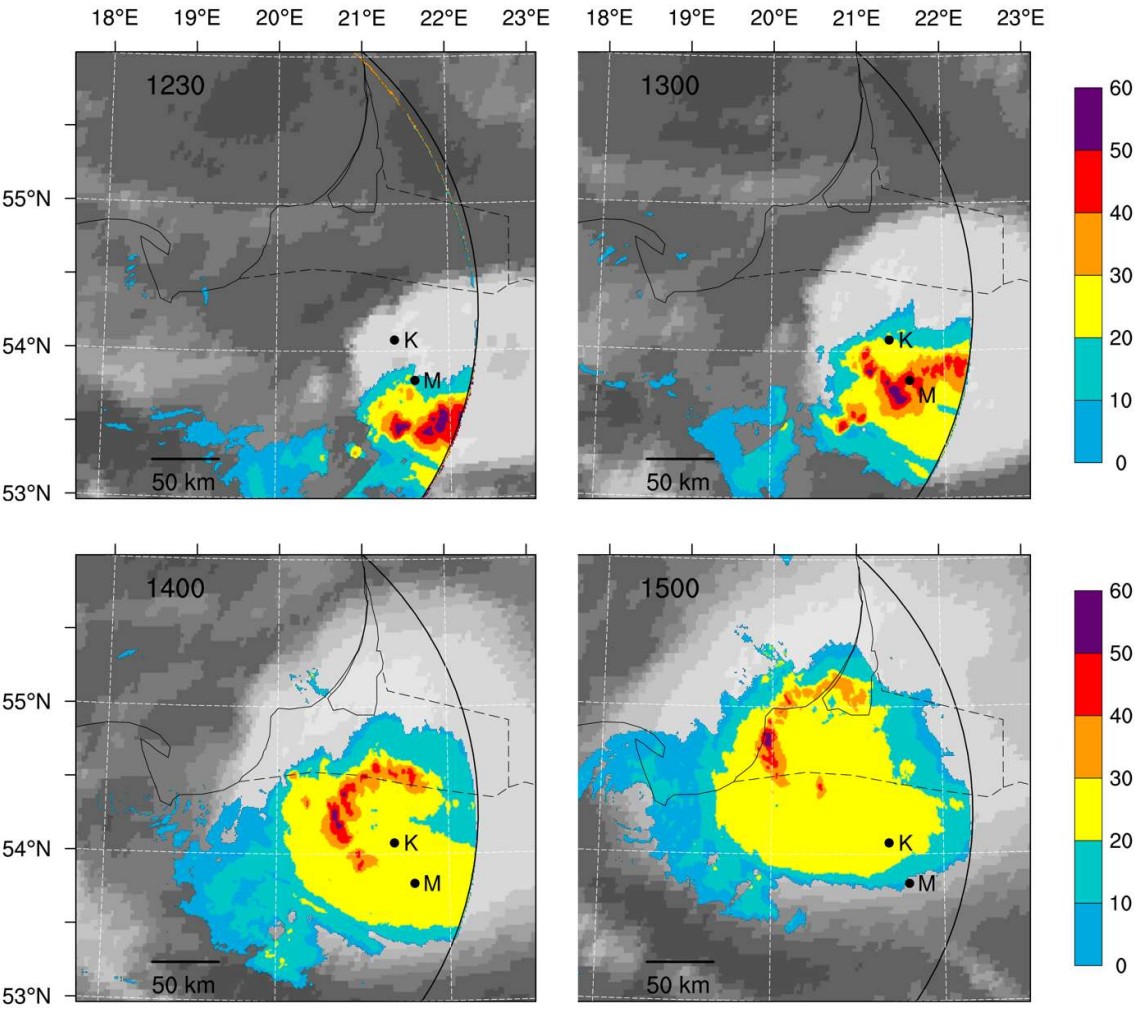

**Figure 5: Radar reflectivity at 6.0 km altitude AMSL (color; dBZ) overlaid over the Meteosat 7.3 µm channel imaging (high/low upper-to-mid tropospheric water vapor concentrations marked with bright/dark colors; EUMETSAT). Black dots denote the locations of Kętrzyn (K) and Mikołajki (M).**

**2.4 Surface observations of the system passage**

The Mikołajki WS recorded the convective system approach at 12:50, reporting very heavy rain shower and severe wind gusts (35 m s$^{-1}$). Hail was observed between 12:58 and 13:01. The total surface precipitation reached 24.6 mm. The 2-m temperature (T) dropped from 27.6 to 16.0°C, and 2-m $T_d$ dropped from 20.2 to 15.5°C. Similar surface observations related to the system's passage were recorded at the Kętrzyn WS 20 minutes later, except for 245 the lack of hail and with only 3.6 mm total precipitation. The strongest 10-m wind gusts reached 30 m s$^{-1}$ (at 13:10), the 2-m temperature dropped from 26.6 to 17.7°C, and the dewpoint temperature dropped from 19.7 to 16.4°C.

**3 Data, tools and initial conditions for the numerical simulation of the convective system**

**3.1 Data and tools**

Reconstruction of atmospheric environmental conditions is based on ECMWF ERA5 reanalysis. It applies 4D-VAR-ensemble 12-hour-window global DA of satellite, aeronautic, radiosonde, and selected surface observations, and an advanced land data assimilation system, coupled with IFS short-term global forecasts (Hersbach et al., 2020). We augment global ERA5 reanalysis with mesoscale DA following the rationale discussed by Gustafsson et al. (2018). Our standard tool is the operational nudging (dynamic relaxation, see Anthes, 1974; Davies and Turner, 1977) of the COSMO Consortium (Schraff, 1997). The scheme works as a non-linear 4-dimensional DA with a flexible time window. The scheme's design prevents the destruction of large-scale balances (e.g., by using sufficiently large time scales) and ensures balances within implemented observations (see extended discussion and technical details in Schraff and Hess, 2021). It has been successfully implemented for operational NWP by COSMO members (Baldauf et al., 2011) also for the assimilation of radar data with latent heat nudging (Stephan et al., 2008). It is also widely used for non-operational studies (e.g., Kienast-Sjögren et al., 2015; Bach et al., 2016; Wilhelm et al., 2023). Nudging methods are also used within ACCORD (Sass and Petersen, 2002) and WRF communities (e.g., Chen et al., 2018).

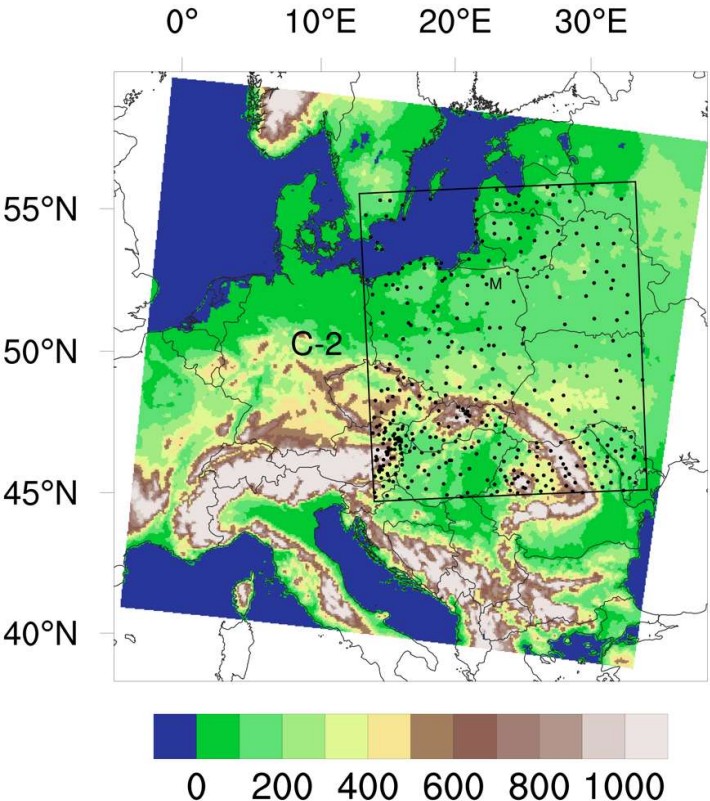

**Figure 6: The COSMO computational domains: larger C-7 (with colored orography in m) and C-2 (black rectangle). Black dots show positions of assimilated weather stations including Mikołajki (M).**

The main prognostic tool is an operational convective-scale NWP COSMO model (Baldauf et al., 2011) version 5.01 applying a 2.2 km horizontal grid spacing (C-2) implemented in the Institute of Meteorology and Water Management – National Research Institute in Poland. In the vertical, it uses 60 levels between the surface and 22.7 km AMSL, and a horizontal domain of 500 x 560 grid points (Fig. 5). Its IC/BC are dynamically downscaled from ERA5 data using the int2lm preprocessor and the convection-parameterizing COSMO 5.01 model with the

horizontal grid spacing of 7.0 km (C-7). These IC/BC include three-dimensional wind velocity and thermodynamic variables with hydrometeors. BC are updated every 30 minutes. The model applies the same vertical structure but a horizontally wider domain of 310 x 310 grid points (Fig. 6).

C-2 and C-7 apply a single-moment cloud microphysical scheme with prognostic ice and snow (Reinhardt and Seifert, 2006) and with additional prognostic graupel for C-2, the multilevel TERRA soil model (Heise et al., 2003)

and 2.5-moment turbulence scheme with an advection of turbulent kinetic energy (Mellor and Yamada, 1982; Raschendorfer, 2001), along with a variant of the statistical cloud scheme of Sommeria and Deardorf (1977; see also Doms et al., 2021). The shallow and deep convection are parameterized (Tiedtke, 1989; Doms et al., 2021) in C-7; there is no convection parameterization in C-2, except a sensitivity experiment mentioned at the end of Section 4.1. The Ritter-Geleyn radiation scheme (Ritter and Geleyn, 1992) provides radiative tendencies updated

every 15 minutes in C-7 and every 6 minutes in C-2.

The orography and surface parameters are derived independently for each resolution using the COSMO EXTPAR software (Asensio et al., 2020) and the leading databases for orography (ASTER; NASA/METI/AIST/Japan Spacesystems and U.S./Japan ASTER Science Team, 2019), land use (GlobCover; ESA GlobCover 2009 Project, 2010), soil type (HWSD; FAO/IIASA/ISRIC/ISS-CAS/JRC, 2012), and surface albedo (MODIS; Schaaf and

Wang, 2015).

Configurations of the COSMO model and of the numerical experiments performed within the current study are summarized in Table 1.

| Models | Description |
|---|---|
| C-7 | $\Delta x$=7 km, 60 levels, IC and BC from ERA5 (except E7-C and E7-M), CP on |
| C-2 | $\Delta x$=2.2 km, 60 levels, IC and BC from C-7, CP off |

| Experiments | Description |
|---|---|
| E7-A | uses C-7, start 21. 07. 2007, 00:00, no further modifications |
| E7-B | uses C-7, start 20. 07. 2007, 00:00, nudging of soil observations |
| E7-C | uses C-7, start 21. 07. 2007, 00:00, soil IC from E7-B, atmospheric IC and BC from ERA5, nudging of soil observations |
| E7-M | Like E7-C, with additional atmospheric IC modification for low-to-mid tropospheric wind alteration |
| E2-A | uses C-2, start 21. 07. 2007, 00:00, IC and BC from E7-A, no further modifications |
| E2-B | uses C-2, start 21. 07. 2007, 00:00, IC and BC from E7-C, nudging soil and surface observations |
| EX | like E2-B, with additional surface heat fluxes correction |
| EX0 to EX8 | like EX, but 9-member ensemble driven by additional SCI |
| EM | like EX, but atmospheric IC and BC from E7-M |
| EM0 to EM8 | like EM, but 9-member ensemble driven by additional SCI |

**Table 1: Configurations of the COSMO model and of the numerical experiments performed within the current study; $\Delta x$ stands for horizontal grid size, CP for convection parameterization and SCI for stochastic convection initiation.**

**3.2 Initial conditions for convective-scale simulations and a correction of convective boundary layer characteristics**

A successful numerical reconstruction of a severe convective system requires realistic environmental conditions supporting the system's development. That includes the temperature and humidity profiles within the atmospheric

boundary layer (ABL) as they influence CAPE and CIN. Within the well-mixed CBL, these profiles can be approximated as functions of 2-m T and $T_d$ (McGinley, 1986) which makes the realistic evolutions of 2-m T and $T_d$ good indicators of realistic temperature and humidity profiles across the CBL, probably except the cloud-base conditions.

However, the C-7 simulation starting at 00:00 on 21 August 2007 with ERA5-based IC and BC (experiment E7-A, see Table 1) systematically underestimates the 2-m T and $T_d$ over northeastern Poland during the whole pre-convective period between 00:00 and 12:00. The RMSE for the period are 1.66°C for T and 0.97°C for $T_d$ for Mikołajki and 2.60°C and 1.01°C for Kętrzyn, respectively. Consequently, the simulated MCIN (around 40 J kg$^{-1}$) and MCAPE (around 2020 J kg$^{-1}$, not shown) over the area at 12:00 are notably overestimated and underestimated, respectively, compared to their observation-based estimates (Fig. 3a). The results of C-2 are similar (experiment E2-A), and the simulation does not develop deep convection in the area (not shown).

### 3.2.1 Nudging of soil and surface observations

We improve these 2-m T and $T_d$ biases in a few steps. First, nudging of routine soil temperature measurements from 12 WSs in northeastern and eastern Poland (Mikołajki, Siedlce, Olsztyn, Białystok, Terespol, Mława, Elbląg, Suwałki, Warszawa-Okęcie, Kozienice, Włodawa, and Lublin) is performed using the simulation E7-B. It starts the previous day (20 August) at 00:00 using C-7 with IC and BC from ERA5. Simulation E7-B provides the IC for soil temperature at 00:00 of 21 August for the corrected C-7 simulation starting at that time (E7-C). The nudging continues within the main C-2 simulation (experiment E2-B) starting at 00:00 on 21 August and lasts until 13:00 with the soil IC taken from E7-B. The atmospheric IC and BC are downscaled from ERA5 using E7-C.

The experiment E2-B additionally performs COSMO nudging of SYNOP observations of 2-m $T_d$, 10-m wind, and surface pressure between 00:00 and 08:00. All available observations within the model domain (Fig. 6) are assimilated including hourly observations from Poland (77 stations) and 3-hourly observations from abroad (225 stations). However, E2-B still incompletely removes the pre-convective 2-m T and $T_d$ errors: RMSE for Mikołajki are 1.61°C and 1.07°C, and for Kętrzyn 1.88°C and 1.24°C, respectively.

### 3.2.2 Modification of surface heat fluxes

As E2-B develops excessive morning cloud cover (compared to satellite observations, not shown), a subsequent experiment EX additionally increases the insolation over northeastern Poland and western Belarus to realistic values characteristic of the cloudless sky, following studies using modified cloud-radiation interactions (e.g., Wu et al., 1998; Harrop et al., 2024 and references therein). The modification is active from 03:30 (approximate sunrise) until 12:00, but is locally turned off if precipitation is detected. The partitioning of the resulting surface heat flux into its sensible and latent components is corrected by altering the initial moisture content of the topmost 0.2 m deep soil layer following, e.g., Yamada (2008) or Gerken et al. (2015). Since the soil moisture measurements are not available, several plausible alternatives were tested to minimize the simulated 2-m T and $T_d$ biases across 13 WSs in northeastern Poland (Białystok, Elbląg, Kętrzyn, Mikołajki, Mława, Olsztyn, Ostrołęka, Siedlce, Suwałki, Terespol) and western Belarus (Baranovichy, Grodno, Lida; see Wójcik, 2021). The finally implemented relative soil moisture corrections vary between 50% (e.g., Olsztyn, Terespol) and -50% (e.g., Kętrzyn, Ostrołęka), see column 5 of table 5.2 in Wójcik (2021).

The applied corrections significantly improve the 2-m T and to a smaller degree $T_d$ forecasts in the pre-convective period over northeastern Poland. RMSE for Mikołajki is reduced to 0.61°C for 2-m T and is 0.73°C for $T_d$, and for Kętrzyn they become 1.32°C and 1.01°C, respectively. That brings the CAPE and CIN of EX close to values estimated from available observations, as discussed in the following section.

## 4 Convection initiation scheme and its impact

### 4.1 Convection forecast without convection initiation scheme

The C-2 simulation with the corrected CBL characteristics, referred to as the EX-forecast, does not develop severe convection over northeastern Poland, despite reproducing the atmospheric environmental conditions in agreement with observation-based estimations from Section 2.2. Local maxima of MCAPE reach locally 2900 and 3300 J kg$^{-1}$ south of Mikołajki at 11:30 and 13:30, respectively, while MCIN immediately south-west of Mikołajki attains values below 10 J kg$^{-1}$ and close to 1 J kg$^{-1}$ already from 11:00 (not shown). The simulation also shows a band of increased low-tropospheric vertical shear (defined as the difference between the wind vectors at 3000 and 100 m AGL) located southwest of Mikołajki (Fig. 7a). Between 12:00 and 14:00, the area of prominent shear of at least 14 m s$^{-1}$ within the band slowly moves toward Mikołajki.

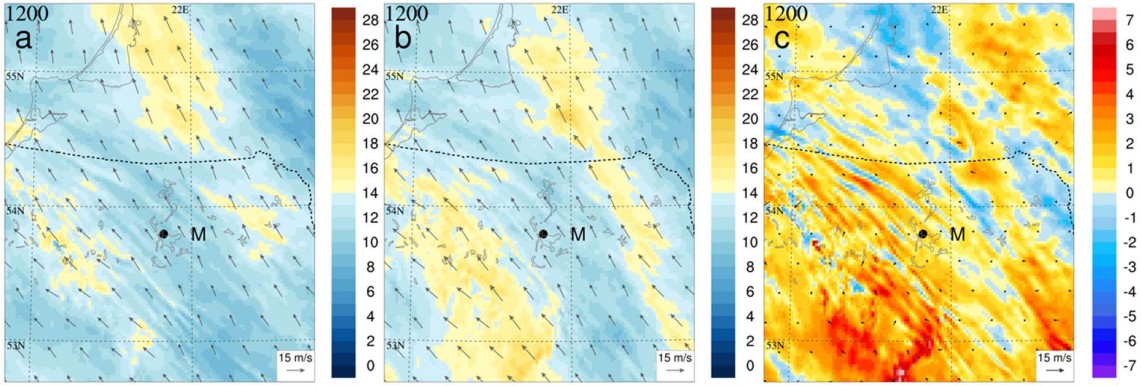

**Figure 7: The 100 to 3000 m vertical wind shear (m s$^{-1}$) in the vicinity of northeastern Poland on 21 August 2007 at 12:00: in EX forecast (a), in EM forecast (b, see section 5), magnitude in color scale, and their difference in color scale (c); black dot show the position of Mikołajki (M).**

The deep convection (defined here as a presence of at least 30 dBZ pseudo-reflectivity at 3000 m AGL) development is noteworthy. It is late, at 12:30, and not in the highest CAPE and low CIN area, but where a belt of increased vertical shear exceeding 15 m s$^{-1}$ coincides with a belt of locally increased MCAPE exceeding 2300 J kg$^{-1}$, about 80 km west of Mikołajki (not shown). That strongly suggests deep convection initiation dynamics according to Peters et al. (2022a; b). They showed that, despite earlier considerations, the high shear environment may promote the process via high-shear-induced dynamic pressure perturbations adjacent to sufficiently developed thermals.

However, the convective gusts (calculated following Brasseur, 2001) do not exceed 15 m s$^{-1}$ until 14:30 and reach only 20 m s$^{-1}$ by 15:00. With the additional lack of bow-shaped convection organization (not shown), the simulation is not successful. Additional measures like the application of shallow convection parameterization (recommended by Doms et al., 2021) did not improve the forecast (convective gusts below 20 m s$^{-1}$ by 15:00). Also, the reduction of asymptotic maximum turbulence mixing length scale tur_len of turbulence parameterization from recommended 150 m (used in our experiments) to 75 m (which is known to help in CI on cost of low-

tropospheric warm temperature bias; Baldauf et al., 2011) marginally improves the forecast with maximum gusts reaching only 19 m s$^{-1}$ until 14:30 and locally 23 m s$^{-1}$ by 15:00 (not shown).

**4.2 Convection initiation scheme**

The model's late deep convection initiation and failure in the development of the severe weather system despite favorable environmental conditions suggest the lack of appropriate CI. Thus, an explicit CI scheme was designed and applied. Its idea is to use near-grid-scale temperature perturbations, possibly resembling those physically developing in CBL, and allow the model to explicitly represent further upscale growth of the perturbations. The

grid-scale perturbations are used in idealized cloud-scale studies (e.g., Tao and Soong, 1986; Grabowski et al., 2006) or in cloud-resolving convection parametrizations (CRCP; Grabowski and Smolarkiewicz, 1999) used within climate models (e.g., Ziemiański et al., 2005). The stochasticity of the method accounts for the limited predictability of the flow at these scales (see Introduction).

At the few-km scales, coinciding with our near-grid scale, large boundary layer thermals with horizontal sizes of

1 to 3 km are observed over land (William and Hacker, 1992, 1993; Marquis et al., 2021), which in the process of convection initiation develop further into convective cells with horizontal sizes of 3 to 5 km (Marquis et al., 2021). Our first experiments used the temperature perturbations representing such large thermals and were applied to single grid columns (horizontal length of 2.2 km). Amplitudes of such perturbations can be estimated following William and Hacker (1992, 1993), showing that (virtual) temperature perturbations of large observed thermals in

relation to their surrounding exceed the convective temperature scale, even by an averaged factor of 2-3 (Fig. 13 of William and Hacker, 1992). Also, for highly heterogeneous underlying surface and moderate geostrophic winds (characteristic for our case), the boundary layer convective temperature scale may be estimated at about 0.4°C (Margairaz et al., 2020). That gives the thermal temperature perturbation in the range of 1°C, which was applied for the model perturbations. Such perturbations, however, did not improve the forecast (not shown).

However, if, besides perturbing a single grid column, the same temperature perturbation is applied also to the four neighbor grid columns, those perturbations substantially impact the CI and are used within this study. The effective horizontal size of those perturbations, taken as the length of a square of the same horizontal surface, is about 2.2 times the grid length (metrically 4.9 km), which coincides with the scale of 3-5-km cells of Marquis et al. (2021). The perturbations, therefore, may be interpreted as representing a near-grid scale flow variability related to such

convective cells, and have sizes smaller than the model's effective resolution of about 7 horizontal grid spacings (Fig. 5 in Ziemiański et al., 2021). Vertically, the perturbations stretch up between the surface and 760 m AGL. The amplitude of the temperature perturbation is drawn from the Gaussian distribution with a mean of 1.25°C and a standard deviation of 0.5°C.

As for assessing the realistic amplitude of temperature perturbations of such convective cells, its lower bound may

be estimated assuming that physically the perturbation results from a dilution of about 3-km-sized thermal over an area of the convective cell, which is about 3 times larger than that of the thermal. That gives the cell's temperature perturbation about 3 times smaller than that of the thermal, which we use in our experiment. If the cells' development also involves merging with neighboring thermals (the process indicated by William and Hacker, 1993; see also Stull, 1988; and Marquis et al., 2021), amplitudes of their temperature perturbation would be larger.

Thus, cautiously, our perturbation amplitudes are stronger by a factor of about 2-3 compared to their realistic values to allow the perturbations to effectively engage with the model dynamics. It may be noted that temperature

perturbations of similar amplitude (1.5°C) were used in the CI context for much larger perturbations, see Zeng et al. (2020) experiment using warm bubbles of about 10 km radius.

The perturbations are activated over eastern and northeastern Poland between 09:30 and 11:30, during six 10-minute intervals. During every such an interval, the perturbations are released at randomly chosen full minutes of the interval and their central columns are located at randomly chosen 4% of the grid columns of the area and are left to grow or decay. Every active interval is followed by a 10-minute pause when the thermals are not released. If the 2-m temperature at the release point is lower than 23°C (a subjective threshold indicating a possible cold pool presence), the thermal is not released. Random configurations of the CI perturbations allow to form an ensemble of 9 forecasts (the size convenient for visualization and alike typically used 10-member ensembles, see Lawson and Gallus 2016 or Hirt et al. 2019), referred to as EX0 to EX8.

It should be noted that this perturbation technique uses only positive temperature perturbations (like the warm bubbles experiment by Zeng et al., 2020). It is therefore biased, as it additionally heats the atmosphere (the averaged effect for 2-m T is 0.2°C at 10:00 and gradually increases to 0.8°C at 12:00 for the comparison between EX0 and EX forecasts in the convective area), breaking the energy conservation principle. That was useful in our experiment as it partly compensates for the negative temperature bias of the EX-forecast for Kętrzyn in that period. The effect for Mikołajki was more ambiguous, as at 12:00 the bias of -0.1°C was modified to 0.5°C. However, potential future applications of the method should be unbiased, e.g., by introducing compensating negative temperature perturbations in the surroundings of the positive temperature perturbations. If the compensating area is sufficient, the compensating perturbations may have absolute values smaller than the positive perturbations and may also be defined in a stochastic way.

**4.3 Impact of CI perturbations on environmental conditions**

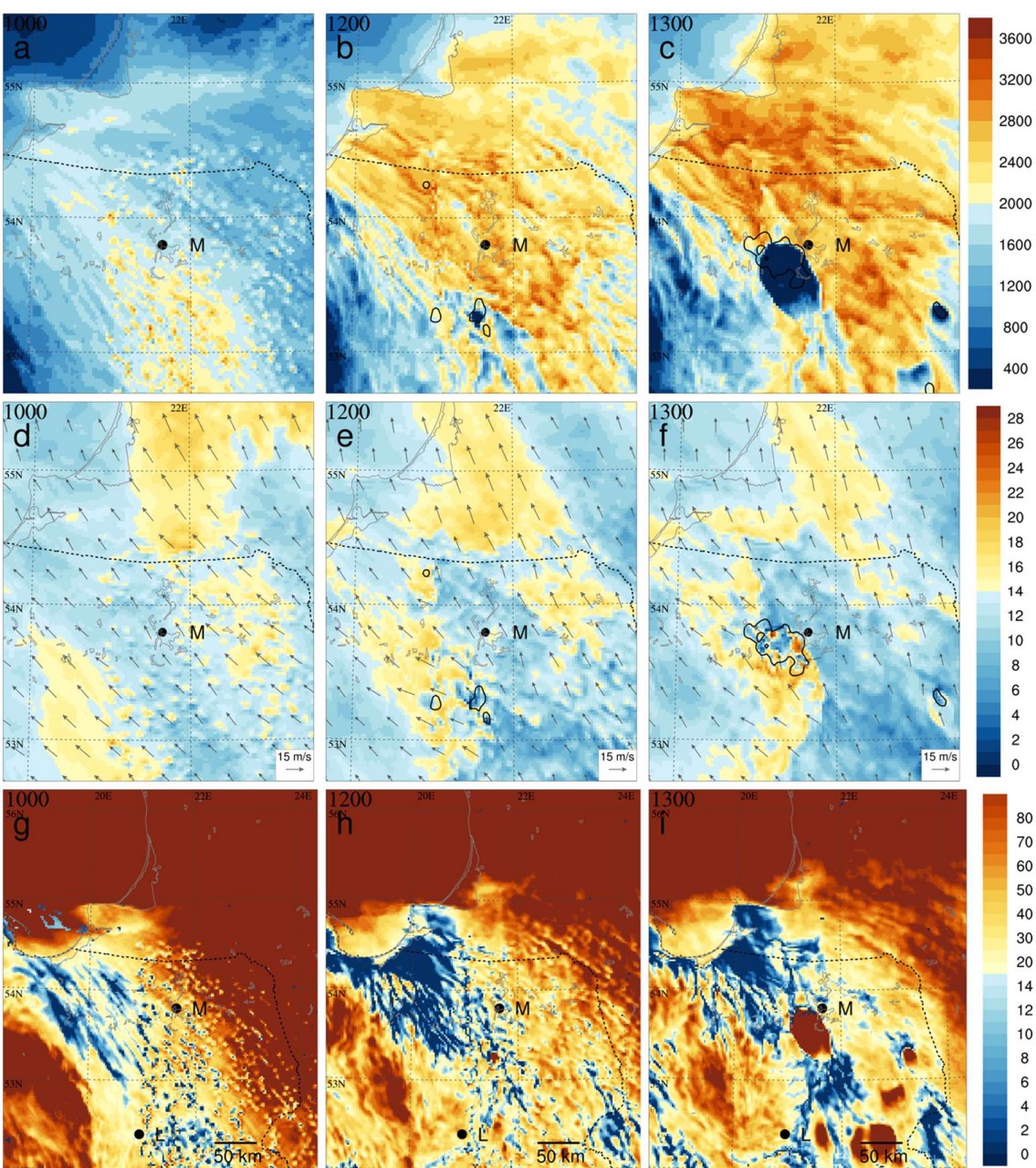

**Figure 8: The environmental conditions in the vicinity of northeastern Poland on 21 August 2007 in EX0 forecast: MCAPE (J kg⁻¹, top row), 100 to 3000 m vertical shear (m s⁻¹, second row), MCIN (J kg⁻¹, third row, note different CIN scales below and above 15 J kg⁻¹), all at 10:00 (left column), 12:00 (middle column) and 13:00 (right column); black contour shows pseudo-reflectivity of 30 dBZ at altitude of 3000 m, black dots show the positions of Mikolajki (M) and Legionowo (L).**

The implementation of CI notably influences the atmospheric environment (see Fig. 8 for the EX0 ensemble member). The increased temperature of the perturbations increases local buoyancy and CAPE and decreases CIN. However, rising thermals force compensating subsidence locally stabilizing the atmosphere. The MCAPE and MCIN spatial distributions in the CI area become grainy at 10:00 with local MCAPE maxima notably larger compared to the undisturbed environment (up to 2800 J kg⁻¹ at 10:00 and 3300 J kg⁻¹ at 11:30 in EX0). MCIN values locally diminish to about 1 J kg⁻¹ at 10:00. Larger-scale regions of high MCAPE (exceeding 2000 J kg⁻¹) in

different ensemble members coincide generally with such regions of the undisturbed EX-forecast (not shown). With time, local MCAPE maxima form filaments and patches, different in location and strength for different ensemble members.

In the whole CI area, the vertical shear distribution (Fig. 8d to 8f) becomes patchy at 10:00 with local maxima increased to 19-20 m s$^{-1}$. Compared to the EX-forecast, the shear patches of at least 14 m s$^{-1}$ within the high-shear band approaching Mikołajki from southwest reach further toward northeast (toward the high-CAPE area). The spatial extent of those patches becomes larger, and the shear maxima stronger, reaching 25 m s$^{-1}$ by 12:30 for some members (EX6 and EX8, not shown). The mechanisms that alter the 3-dimensional wind distribution (and shear) are highly nonlinear and involve convective momentum transport and accelerations imposed by convective pressure perturbations (e.g., Wu and Yanai, 1994; Schlemmer et al., 2017; Dixit et al., 2021). These mechanisms were not analyzed within our study and are beyond the scope of this paper.

## 4.4 Development of deep convection and its cold-pool-driven dynamics

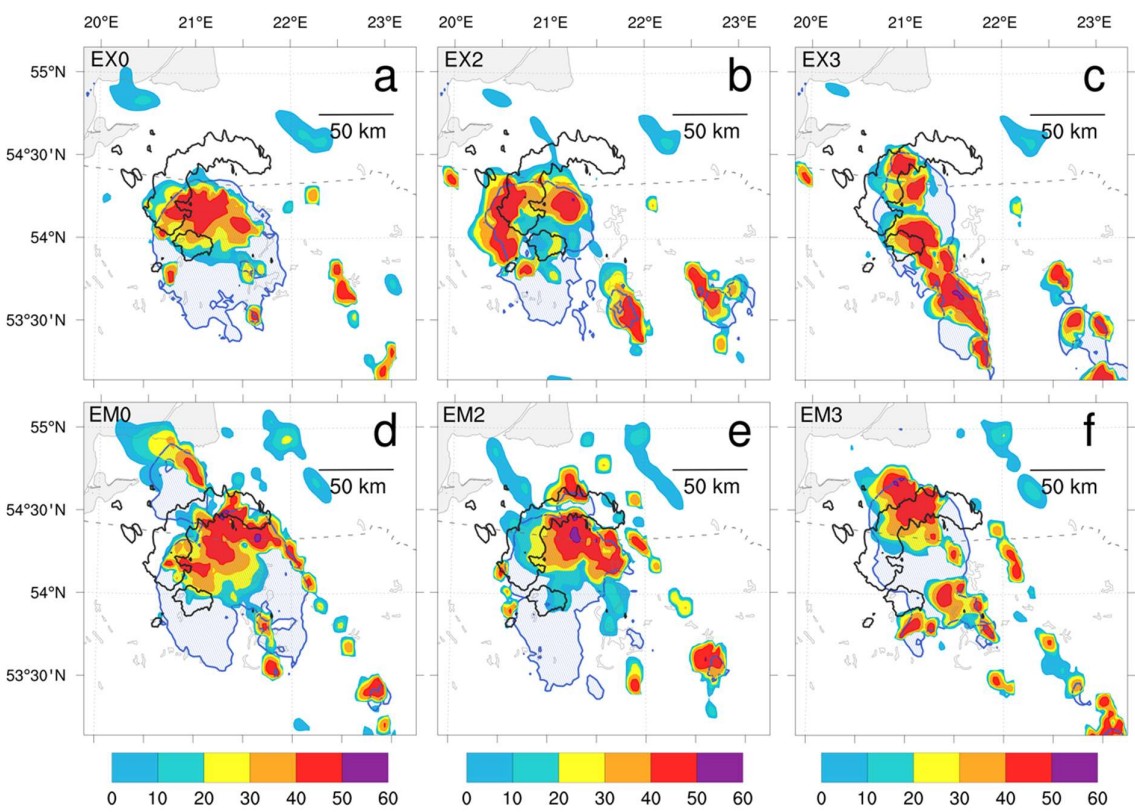

**Figure 9: Simulated pseudo-reflectivity (dBZ, color scale) and cold pool extent (blue shading) for the EX0 (a, strong gusts), EX2 (b, moderate gusts) and EX3 (c, weak gusts) ensemble members at 14:00 (their numbers shown in upper-left corners) with area of the 30 dBZ reflectivity (black contour) of the observed system; in the second row the same information for the analogous members of the EM ensemble (Section 5): EM0 (d), EM2 (e), EM3 (f).**

The first deep convective cells with pseudo-reflectivity reaching 30 dBZ at 3000 m AGL develop between 11:00 and 11:30 (depending on the ensemble member) in the area south, southwest and west of Mikołajki on locally increased CAPE features (not shown). Some of those cells develop further, producing cold pools, sufficiently strong to significantly reduce MCAPE in the cells' area between 12:00 and 13:00 (e.g., Fig. 8c), and severe gusts, later. In those cases, the process begins where the high-shear patches (at least 14 m s$^{-1}$), located within the eastern edge of the high-shear band heading toward Mikołajki from southwest, approach the high CAPE features (not

shown). That strongly suggests the decisive role of high shear via a mechanism discussed by Peters et al. (2022a; b) in the early model-represented development of such deep convection systems.

| Ensemble members | EX0/ EM0 | EX1/ EM1 | EX2/ EM2 | EX3/ EM3 | EX4/ EM4 | EX5/ EM5 | EX6/ EM6 | EX7/ EM7 | EX8/ EM8 | Av. EX/ EM |
|---|---|---|---|---|---|---|---|---|---|---|
| Max gusts by 13:00 | 25/24 | 18/26 | 20/23 | 18/18 | 24/27 | 16/26 | 18/23 | 19/17 | 18/15 | 20/22 |
| Max gusts by14:00 | 30/38 | 30/29 | 25/30 | 25/25 | 29/24 | 26/34 | 30/31 | 23/28 | 33/27 | 28/30 |
| Max 2-m T depr., 13:00 | 7.5/7.7 | 6.6/7.1 | 6.8/6.9 | 5.5/6.1 | 7.7/7.3 | 6.5/8.1 | 6.5/6.8 | 5.2/7.1 | 5.2/5.8 | 6.4/7.0 |
| Max 2-m T depr., 14:00 | 8.4/7.9 | 8.2/8.5 | 8.0/7.9 | 7.2/7.3 | 8.0/7.3 | 7.9/7.9 | 8.8/7.7 | 7.3/6.5 | 8.6/7.6 | 8.0/7.6 |

**Table 2. Maximum gusts (m s⁻¹) at half-hourly periods between 12:30 and 13:00, and between 13:30 and 14:00 UTC, and cold pool amplitudes (°C) at 13:00 and 14:00 UTC for the analogous members of EX (blue, Section 4) and EM (red, Section 5) ensembles with ensemble averages.**

At 13:00, all ensemble members produce clusters of deep convective cells in the vicinity of the observed system (not shown). They are not yet organized into bow-shaped systems but have developed significant cold pools (see

Table 2; the cold pools' temperature depression is calculated for the 2-m T relative to the EX-forecast).

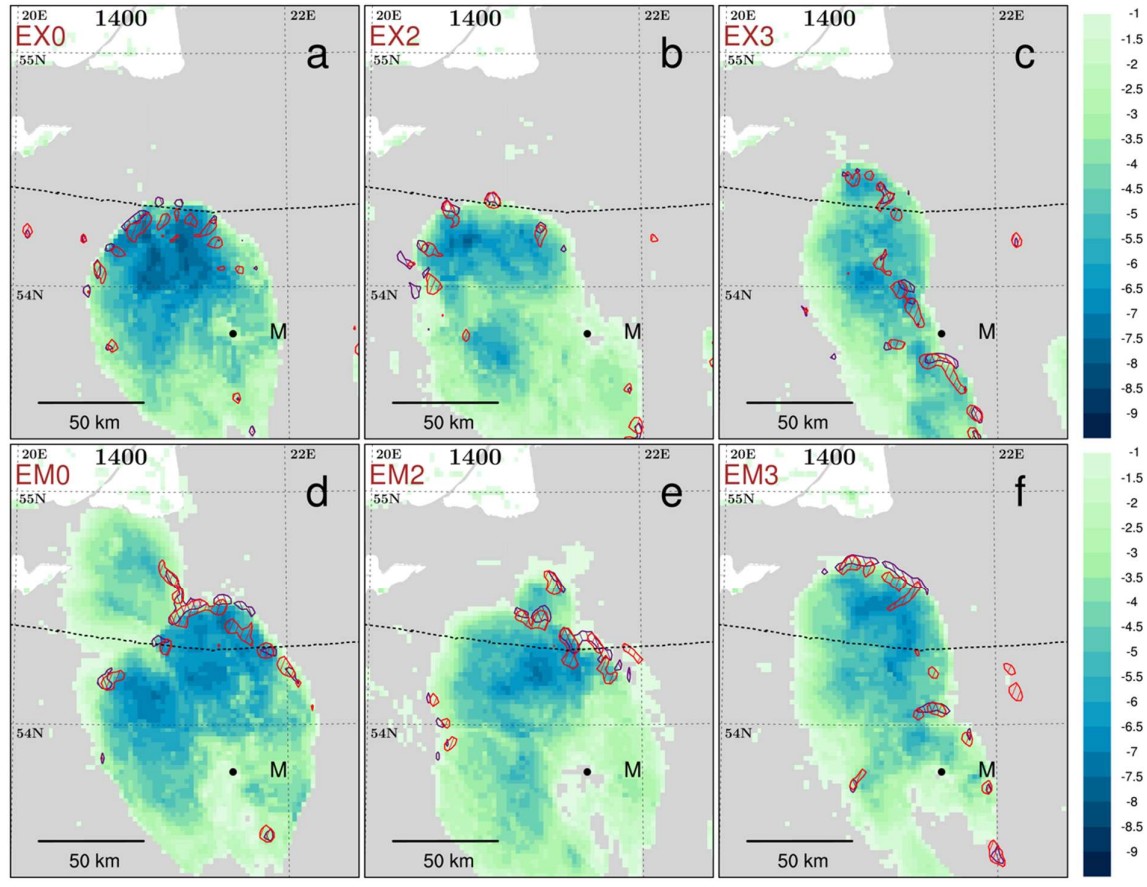

**Figure 10: 2-m temperature depression (color, °C; see text for definition) and the isolines of 5 m s⁻¹ vertical wind at the altitude of 3 km AMSL (violet hatching) and 6 km AMSL (red hatching) for EX0, EX2 and EX3 ensemble members (numbers in upper-left corners) at 14:00; black dot for the position of Mikołajki (M); in the second row the same**
**information for the analogous members of the EM ensemble (Section 5): EM0 (d), EM2 (e), EM3 (f).**

By 14:00, the cold pools intensify (Table 2), and all ensemble members feature the convective clusters with cells located in the vicinity of the cold pools' leading edge, for most of them (EX0, EX2, EX4, EX6, EX7, EX8) along a bow-shaped line (Fig. 9a to 9c for EX0 representing ensemble members with the strongest gusts, EX2

representing members with moderate gusts, and EX3 representing members with the weakest gusts, other ensemble members not shown). The latter suggests the presence of cold-pool-driven dynamics in the development of the clusters, even though the cells tend to be separated from each other.

That is verified analyzing the locations of convective updrafts, stronger than 5 m s⁻¹, relative to the cold pools positions, the latter identified by the temperature depression pattern. The analysis indicates that all ensemble forecasts feature strong convective updrafts located along the cold pool's leading edge (Fig. 10a to 10c for the ensemble members EX0, EX2, EX3, other ensemble members not shown), thus confirming that the convective clusters of all ensemble members are subjected to the cold-pool-driven dynamics despite the individual updrafts being isolated and not forming a compact linear structure. Further confirmation comes from the subsequent evolution of all the systems (not shown) that keep developing strong updrafts (which tend to increase in number and spatial extent) on the cold pools' leading edges. At 14:00, the cold pools' leading edges and the related convective clusters are located relatively close to the actual system's position but –except EX3– stay clearly behind it (Fig. 9a to 9c for EX0, EX2, and EX3).

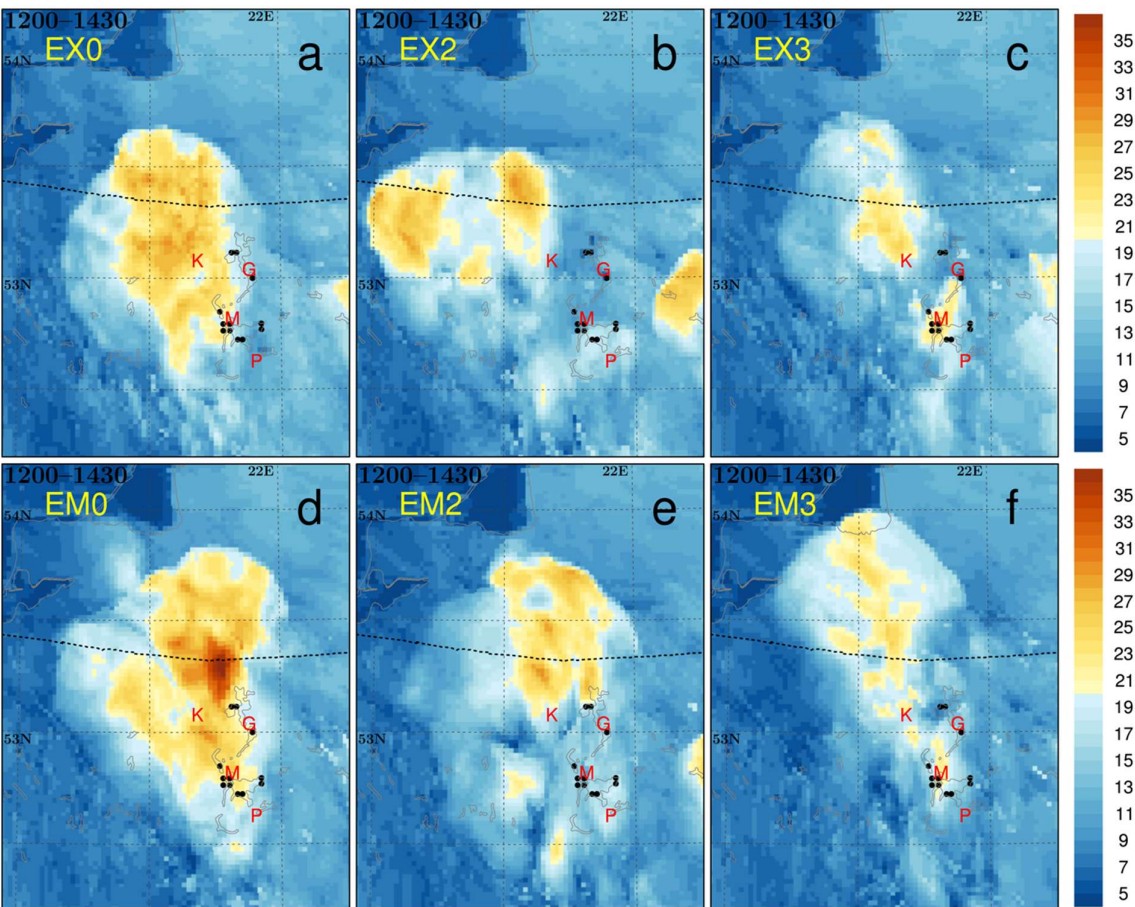

**Figure 11: Spatial distribution of maximum 10-m wind gusts (m s⁻¹) in the period between 12:00 and 14:30 for the EX0, EX2 and EX3 members of the EX-ensemble with positions of towns with damaging winds reports (based on press releases) marked by red capital letters and positions of death reports marked by black dots; in the second row the same information for the analogous members of the EM ensemble (Section 5): EM0 (d), EM2 (e), EM3 (f).**

By 14:00, four ensemble members produce maximum gusts between 23 and 26 m s⁻¹ and five members between 29 and 33 m s⁻¹ (Table 2). It is interesting to compare the spatial distribution of maximum gusts between 12:00 and 14:30 with the distribution of available damaging wind reports (Figure 11a to 11c for EX0, EX2, and EX3, other ensemble members not shown). While none of the forecasted areas of gusts of at least 20 m s⁻¹ covers the whole

area of damaging wind reports, most of these simulated areas are relatively close or cover at least part of the damaging wind area (EX0, EX1, EX3, EX4, EX5, and EX6). There is an overall tendency to develop strong gusts further west and north compared to their actual position. That is at least partly related to a model's delay in the gusts' development. Overall, however, the implementation of the CI scheme radically improves the forecast, making it comparable to the observed system.

## 5 Increased-shear experiment

There are no observations directly verifying the lower-to-mid-tropospheric wind velocities simulated within the high-wind band over the Mikołajki area during the convective development. However, the simulated slower propagation of the convective cluster between 13:00 and 14:00 (see previous section) allows us to hypothesize that the velocities are underestimated in the model simulations (e.g., Weisman and Klemp, 1986). One may expect that the thermal wind relation played a role in the development of those winds (850-500 hPa thickness evolution, not shown). Such a conjecture suggests an experiment to increase wind velocity within the band (and thus increase the vertical shear) and tests the impact on the simulated convective system's strength, morphology, and timing. The wind is strengthened by increasing its balanced component via local strengthening of the low-to-mid-tropospheric horizontal temperature gradient.

### 5.1 Shear modification technique

The experiment is set up via appropriate modifications of lower-to-mid-tropospheric temperatures in the source area of air that departs at 00:00 and reaches the colder/warmer (western/eastern) side of the high-wind band over Mikołajki around 12:00. The source areas are defined using trajectory analysis (Fig. 12; the trajectories are calculated with Lagranto software following Sprenger and Wernli, 2015), which also shows that the band coincides with a local convergence zone indicated by the converging trajectories. The modifications are implemented within the E7-M experiment that provided the IC and BC for the convective-scale experiment with increased shear, referred to as EM. To keep the experiment within realistic bounds, the applied temperature modifications, albeit arbitrary, are bounded by available upper-air observations.

The temperature modifications for the eastern side of the increased temperature gradient are performed in the source area of trajectories arriving east of Mikołajki at 12:00, located near and south of the Shepetivka upper-air station (Fig. 12 for the applied temperature perturbations). The modification area is contained between 900 and 520 hPa and has the form of an ellipse having foci located at 27.05°E, 50.18°N, and 28.10°E, 47.25°N; the semi-major axis equals 201 km. The temperature is increased to measurements from Shepetivka sounding at 00:00 and is modified to be horizontally uniform over the area. Additionally, the humidity in the area is reduced to values close to those observed at Shepetivka (not shown) to avoid triggering spurious morning convection.

The modifications for the western side of the increased gradient zone are performed in the source area of trajectories arriving near the Legionowo upper-air station at 12:00 (Fig. 12). The modification area is contained between 830 and 540 hPa and has a form of ellipse with foci located at 19.60°E, 48.95°N, and 20.90°E, 45.85°N; and its semi-major axis equals 225 km. The modified temperature is horizontally uniform. There is no representative upper-air station in the area, so the temperature is lowered to make the simulated temperature profile at Legionowo at 12:00 closer to the observed one. With modification amplitudes slightly stronger compared to those around Shepetivka (up to -3 °C compared to 2.5 °C), the Legionowo temperature in the lower-to-mid

troposphere was reduced on average by 0.7 °C and is closer to observations (not shown). The model quickly
recovers the balance and increases the horizontal temperature gradient in lower-to-mid troposphere over the severe
convection area around noon (not shown).

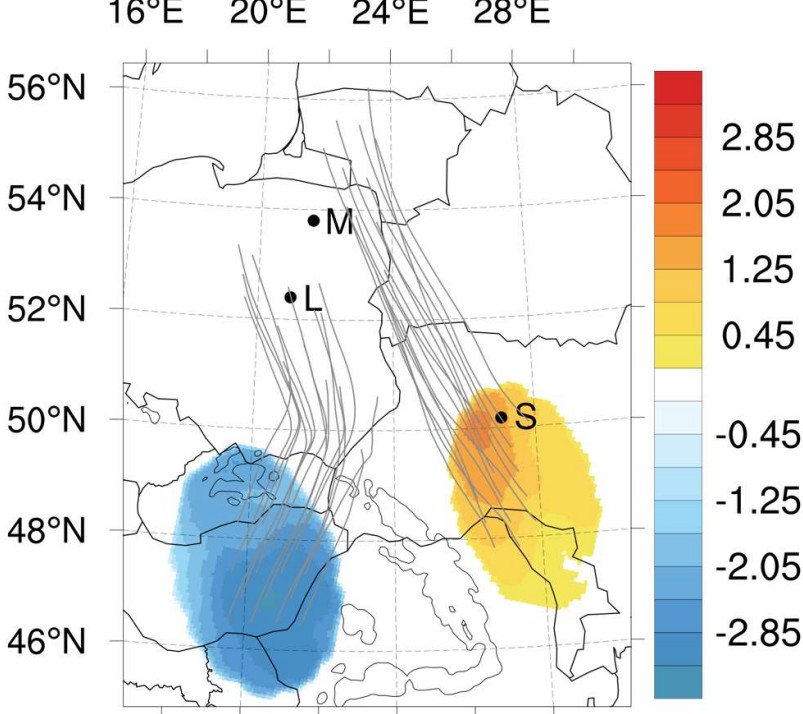

**Figure 12: The temperature perturbations (°C, color) imposed at 00:00 together with forward parcel trajectories between 00:00 and 12:00 of 21 August 2007 for negative temperature modification at 650 hPa and for positive temperature modification at 700 hPa. M, L and S mark locations of Mikolajki, Legionowo and Shepetivka, respectively.**

As expected, the modifications mainly influence the vertical shear amplitude in the EM-forecast while preserving its overall mesoscale pattern over northeastern Poland between 10:00 and 14:00. In the EM-forecast, as in the basic EX-forecast, the shear slowly weakens with time but remains stronger by 2-3 m s$^{-1}$ in most of the area near Mikołajki, locally even by about 5 m s$^{-1}$ by 12:00 and by about 7 m s$^{-1}$ at 13:00, compared to the EX-forecast (Fig.
7b, c for 12:00). The thermodynamic conditions are also alike, including MCIN and MCAPE overall patterns (not shown).

## 5.2 Convection forecasts with increased vertical shear

Overall, the increase in shear alone did not improve the severe weather forecast. In the EM-forecast without stochastic CI, deep convection develops at 13:00 about 40 km south of Mikołajki, in the area where a narrow belt
of shear reaching 17 m s$^{-1}$ catches up with a narrow belt of MCAPE exceeding 3000 J kg$^{-1}$ (not shown), in agreement with the dynamics discussed by Peters et al. (2022a; b). Compared to the EX-forecast, the maximum convective gusts are weaker (down to about 15 m s$^{-1}$ by 15:00) despite convection developing within the area with accessible stronger CAPE and shear.

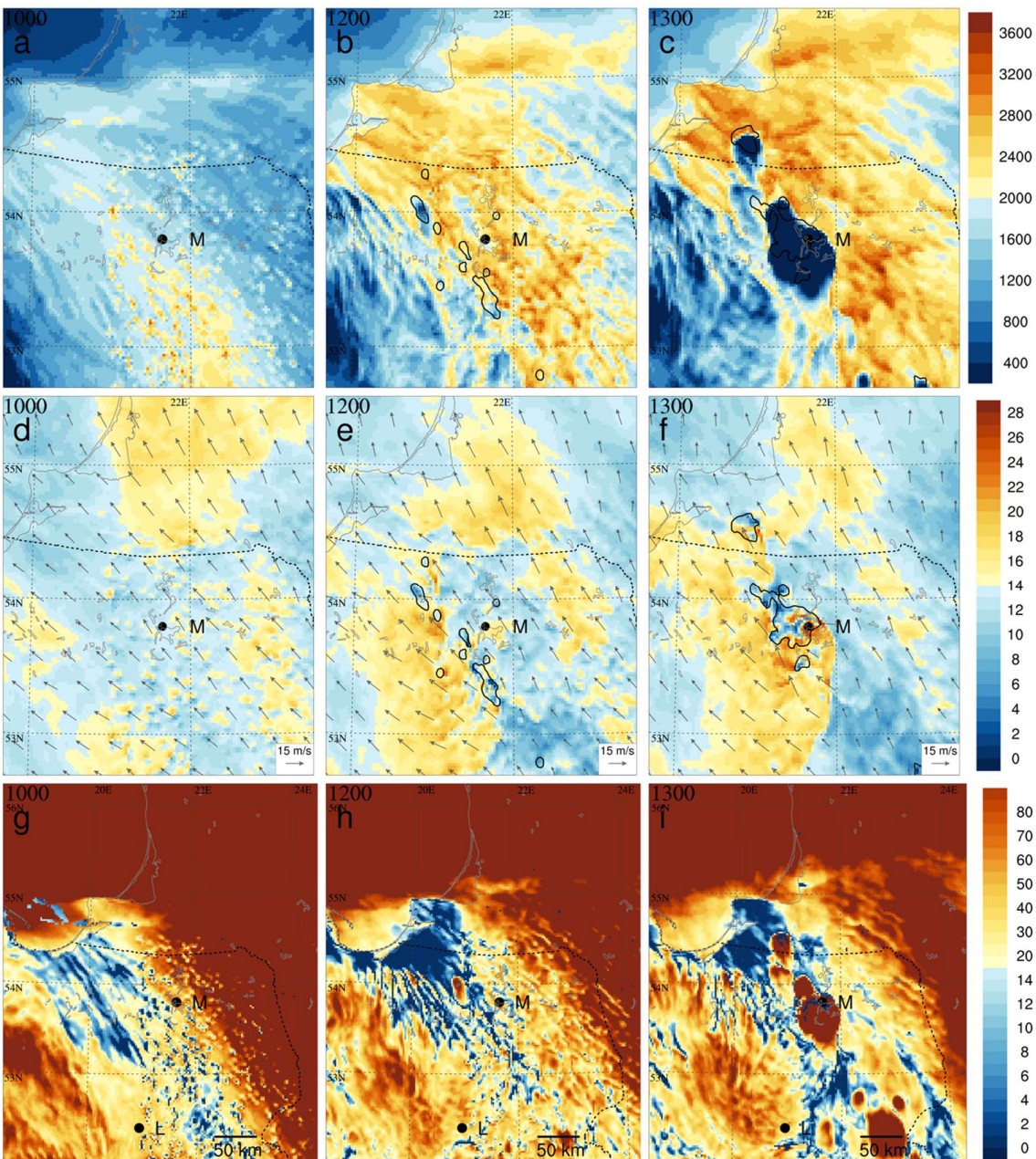

    **Figure 13: As Fig. 8 but for EM0 ensemble member.**

Further implementation of the CI scheme within the EM-forecast produces a new EM-ensemble and decisively changes the convection forecast. The ensemble members EM0 to EM8 are generated using the saved configurations of the stochastic CI used to generate ensemble members EX0 to EX8. As in the EX-ensemble, the implementation of CI alters the distribution of environmental parameters, making their small-scale structure grainy or patchy (see Fig. 13 for the EM0 member, other ensemble members not shown). Compared to the analogous EX-ensemble members, the local shear maxima are stronger by 1-3 m s$^{-1}$ already at 10:00. They quickly increase and reach 24-25 m s$^{-1}$ by 11:00 in most of the ensemble members. Values of local MCIN minima (around 1 J kg$^{-1}$) and local MCAPE maxima are similar in both ensembles. Small-scale structures of MCIN and shear are at first similar for the analogous members of both ensembles but diverge from about 11:00 for shear and 12:00 for MCIN.

Deep convective cells (pseudo-reflectivity reaching 30 dBZ at 3000 m AGL) develop earlier (between 10:30 and 11:00) in the EM-ensemble (not shown). Intensive developments take place where the locally increased shear patches (at least 14 m s$^{-1}$) approach the increased CAPE features (not shown), as in the EX-ensemble. Already at 13:00, all EM-ensemble members produce clusters of deep convective cells with deep cold pools (Table 2; the cold pools' temperature depression is calculated for the 2-m T relative to the EM-forecast) in the close vicinity of

the actual system's position (not shown), but the cells are not organized into bow-shaped systems.

By 14:00, all ensemble members feature convective clusters with cells located in the vicinity of the cold pool leading edges, and all but EM3 along a bow-shaped line (Fig. 9d to 9f, for EM0 representing ensemble members with strongest gusts, EM2 representing members with moderate gusts, and EM3 representing members with weakest gusts, other ensemble members not shown). Compared to the EX-ensemble, there are more such ensemble

members, and cold pools tend to be more widespread. Moreover, the convective clusters are located closer to the position of the observed system, indicating an improvement in the system's propagation speed.

The analysis of convective updrafts' positions (Fig. 10d to 10f, for the ensemble members EM0, EM2, EM3, other ensemble members not shown), indicates that all EM-ensemble members feature strong convective updrafts organized along the cold pool's leading edges, confirming the cold-pool-driven dynamics of these convective

systems. The updrafts tend to be more widespread compared to the EX-ensemble members, some forming lines of 10-km-length scale. For some of the ensemble members, the cold pools attain a more complicated dual structure, with the updrafts forming at the leading edges of both parts (EM2, EM3, EM4, EM5, EM7, EM8). Later, for all ensemble members, the convective systems keep developing strong updrafts on the cold pools' leading edges, confirming the persistence of the cold-pool-driven dynamics (not shown).

The maximum gusts in the EM ensemble are stronger compared to the EX-ensemble (Table 2), with four ensemble members producing maximum gusts between 25 and 28 m s$^{-1}$ and five members between 29 and 38 m s$^{-1}$. Also, the spatial distributions of maximum gusts between 12:00 and 14:30 tend to be closer to reality (Fig. 11d to 11f, for the ensemble members EM0, EM2, EM3, other ensemble members not shown). EM0 and, to a large degree, EM5 have gusts of at least 20 m s$^{-1}$ covering the area of damaging wind reports, and such gusts of all members but

EM2 cover at least some of that area. There is still a tendency to forecast strong gusts further west and north (at least partly related to a slower spin-up of simulated convective processes) compared to their actual position.

The forecast ensemble using the CI responds well to the increase of low-to-mid tropospheric wind and shear via increasing the maximum gusts of the ensemble and convective system propagation speed while making the spatial structure of convective cells and updrafts more bow-echo-like.

**6 Rear inflow jet in the EM0 forecast**

The model's problems with the development of a bow-like cloud structure (related to the model limitations discussed in the Introduction) indicate that the model may also struggle with representation of other physical aspects of the analyzed convective development, including the RIJ formation. Therefore, a brief analysis of how the model-represented convective processes influence the low-to-mid tropospheric flow is performed. The EM0

forecast is chosen for the purpose because it gives the timing and position of the convective system most closely resembling the observations, and wind gusts in the period between 13:00 and 14:00 are the strongest (38 m s$^{-1}$). It may be assumed that the forecast gives the best approximation to the real development. Figure 14 shows the 700-hPa wind in the vicinity of the developing convective cluster. The figure indicates that there is no significant wind

increase close to the convective cells at 12:00. At 13:00, the wind speed increases in the rear part of the strong

echo area, reaching 29 m s⁻¹ in a small area above Mikołajki. At 14:00, the local area of the strongest wind reaching

33 m s⁻¹ increases forming two patches. Only at 15:00, a relatively large and compact area of strong wind reaching

36 m s⁻¹ is seen, suggesting a presence of a well-developed RIJ.

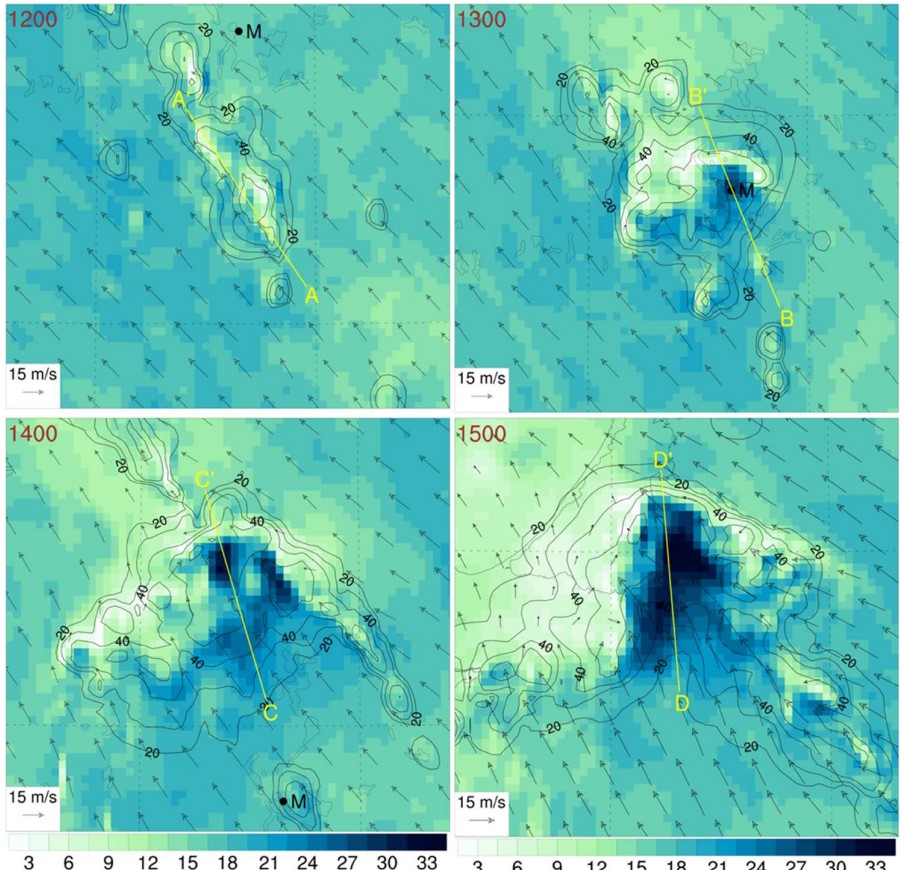

**Figure 14: Wind speed (color scale in m s⁻¹) and wind vectors at 700 hPa in the vicinity of the convective cluster with**
**column-maximum pseudo-reflectivity (CMAX, dBZ in black contours), hour in UTC in left upper corners. The black**
**dot shows the location of Mikołajki (M), and the yellow lines show the positions of cross-sections; their length is 65 km.**

The development of RIJ can be better assessed with Fig. 15 that shows vertical cross-sections of wind perturbations

through the developing convective cluster. The flow perturbations are calculated relative to the EM-forecast

(running without CI) at the cross-section time and place, and its component is shown along the direction of the

reference wind averaged over the cross-section in 700-500 hPa layer (the direction is between 130°-140° and the

averaged wind velocity diminishes slowly between 12:00 and 15:00 from 16.3 to 13.7 m s⁻¹). Already at 12:00, an

increase of wind speed reaching 4 m s⁻¹ at 5 km altitude is seen behind the convective updraft. At 13:00, a 30-km-

long zone of positive wind speed perturbation forms in the low-to-mid troposphere behind the strong updraft with

a maximum reaching 15 m s⁻¹ at about 3 km altitude, indicating a formation of the RIJ. It may already contribute

to surface gusts as its frontal part is caught in a strong downdraft reaching the near-surface area. At 14:00, the

length of RIJ exceeds 50 km and its velocity perturbation reaches 18.5 m s⁻¹ in the frontal part of the system, also

at about 3 km altitude, where the pressure perturbation exceeds -250 Pa. The middle part of the RIJ, caught in a

strong downdraft reaching the near-surface area, likely contributes to the forecasted very strong surface gusts. At

15:00, the RIJ intensifies across its extensive area with maximum velocity perturbations reaching 20 m s⁻¹ but a

transfer of its momentum to the near-surface area seems to be weaker.

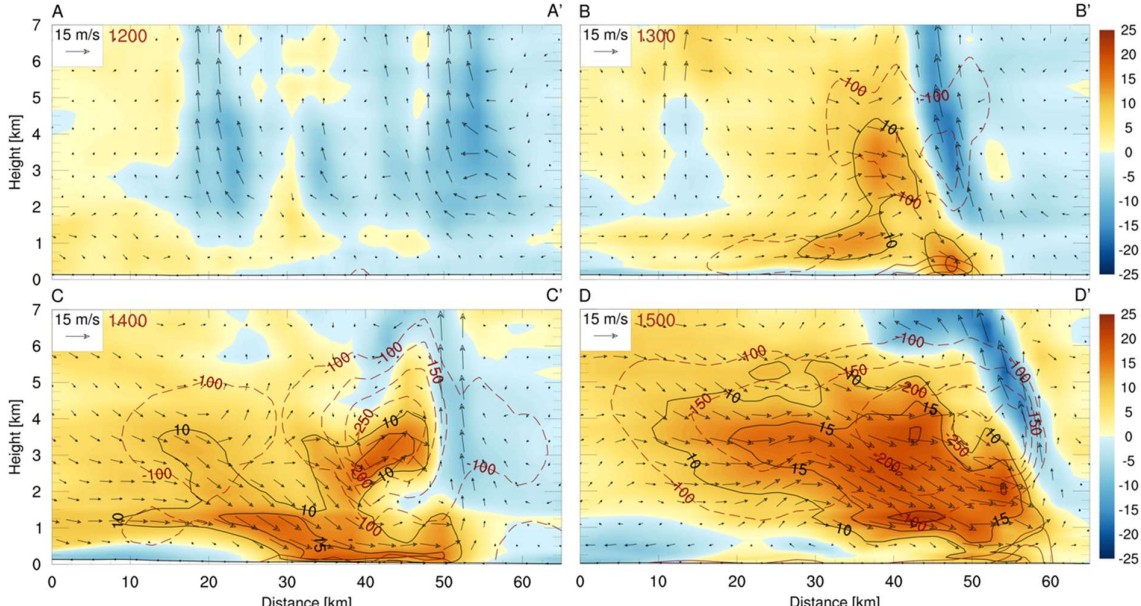

**Figure 15:** Vertical cross-sections through horizontal wind perturbation relative to reference EM-forecast; its component along the direction of the reference wind averaged over the cross-section in 700-500 hPa layer is shown (color scale and black isotachs at 10, 15 and 20 m s⁻¹), black vectors show the horizontal and vertical perturbations of flow velocity, red lines show pressure perturbations at plus/minus 100, 150, 200, 250, 300 Pa (negative contours are dashed). Hours in UTC are in the upper-left corners, capital letters on top of the plates indicate start and end points of the cross-sections shown in Fig. 13.

The analysis shows a lack of significant model problems in the RIJ simulation: the strong RIJ forms already at the very early stage of the convective system development, that is at 13:00. Moreover, the resulting convective circulation allows the RIJ to influence the magnitude of the 10-m wind and gusts already at that early development stage (and later). The model limitations do not significantly impair its ability to simulate the RIJ formation. The simulation strongly suggests that an early developing RIJ may notably influence the catastrophic strength of convective gusts actually observed at the early stage of the developing bow echo near 13:00.

## 7 Summary and conclusions

The paper presents a numerical simulation of a rapidly-developing, fast-propagating, severe convective meso-β-scale bow echo system that formed over northeastern Poland on 21 August 2007. The system was weakly forced in the sense of the lack of the omega equation forcing for strong synoptic-scale ascent or nearby frontal surfaces. However, it developed in the vicinity of a local convergence zone (Section 5.1). The system produced maximum surface wind gusts of 35 m s⁻¹, and caused significant property damage and 12 fatalities.

As discussed in the Introduction, NWP models are – by design – not able to fully represent the physical and dynamical processes acting across the entire range of spatial and temporal scales to develop organized convection. Indeed, we demonstrate that the operational convective-scale COSMO model with a horizontal grid spacing of 2.2 km used for our study encounters significant problems with numerical reconstruction of the event, even with favorable atmospheric environmental conditions and additional application of shallow convection parameterization or reduced maximum turbulence mixing length tur_len. However, an implementation of a near-grid-scale stochastic convection initiation (CI) scheme, using temperature perturbations reminiscent of about 5-km-size convective cells, allows the relatively coarse-grid and under-resolving model (see the Introduction) to explicitly reconstruct convective development. With a provision of realistic environmental conditions and the

stochastic CI mechanism implemented within a 9-member ensemble, the model can reconstruct the timing, place, and severity (maximum gusts exceeding 30 m s$^{-1}$) of the fast-developing and meso-$\beta$-scale convective system.

The model realistically reconstructs the system's cold-pool-driven dynamics organized by strong updrafts at the leading edge of its cold pool. The model is able to develop an RIJ of realistic extent and magnitude in early stages of the system evolution. The simulations respond well to the increase of low-to-mid tropospheric winds and vertical shear: the maximum gusts of the ensemble and the system's propagation speed increase, while the structure of convective updrafts at the cold pools' leading edge is improved. However, the model notably delays the development of the strongest gusts (by almost an hour) and struggles with the formation of a continuous convective line. In contrast to the observations, simulated convective cells that develop on the cold pool leading edge do not merge sufficiently early and tend to stay as isolated entities. The study demonstrates that, despite these drawbacks, the convective-scale NWP models have a valuable potential in the prediction of such severe convective systems that have a high social impact.

As for the limitations of our experiment setup, we already discussed its temperature bias and perturbation amplitudes likely stronger (2-3 times) than realistic. We also did not optimize the method in terms of spatial and temporal density, nor the final amplitudes and shapes of the perturbations. We think that the proposed CI scheme may be useful for deep convection studies. Such a perturbation strategy, corrected for its temperature bias, may still be of interest for the NWP applications because it allows continuous stirring of near-grid-scale variability in a way that accounts for the model's dissipative properties. We consider it reasonable that future applications of such a CI procedure will deliberately use near-grid-scale but overestimated temperature perturbations.

Also, the size of our ensemble, while typical for other similar studies, can be regarded as small because many considerations suggest that the optimum size of the prognostic convective-scale ensembles is a few orders of magnitude larger (e.g., Uboldi and Trevisan, 2015; Bannister et al., 2017; Necker et al., 2020; Craig et al., 2022). However, our process-based approach linked with a restoration of likely mesoscale environmental conditions gives an ensemble that is reliable in the sense that all the ensemble members predict the development of a cold-pool-driven convective system, and such a system was observed there. The system's forecasts vary basically in its intensity, measured especially by the strength of surface gusts. That suggests that the applied perturbation strategy addresses to a considerable degree the uncertainty of the analyzed convective process related to small (near-grid) atmospheric scales and that these scales have a decisive impact on the process. It also suggests that the development of such a convective system was very likely with the observed mesoscale conditions and the presence of sufficient CI.

The following concluding comments may be formulated:

--- The study confirms the important role of near-grid-scale variability of atmospheric flows for the development of convective systems such as the one analyzed here. Because of the limited model resolution (mostly horizontal but also vertical), current convective-scale NWP models cannot represent such a variability, and the problem may be confronted by perturbation methods like the CI scheme used here.

--- For convective processes, the paradigm of perturbation sizes limited by the model's effective resolution can be relaxed by implementation of near-grid-size perturbations resembling those developing within CBL, at the cost, however, of an increase of their magnitude, likely by a factor of 2-3, to respond to unphysical damping of such perturbations by the numerical model. An interesting avenue for further research opens here as the work of Peters et al. (2022a; b) suggests that there may still exist a *physically* based horizontal lower size limit for such perturbations in high-shear environments typical for severe convection.

--- The temperature-only perturbations lead not only to perturbations of MCAPE and MCIN but also to significant perturbations of low-to-mid tropospheric winds and vertical shear. Those perturbations evolve interacting with the ambient flow and form local extrema exceeding those of the ambient flow.

--- Local maxima of the vertical shear play an important role in model representation of deep convection initiation and in further development toward the convection organization, arguably due to the dynamical mechanisms

analyzed by Peters et al. (2022a; b).

--- The study suggests that using stochastic near-grid-scale perturbations may be of value for improving operational convective-scale NWP via stirring the flow variability at these strongly damped but physically significant scales. That would require substantial further work, ensuring conservation requirements, following, e.g., Berner et al. (2017).

--- Finally, realistic convective case studies using IC/BC derived from global reanalyzes (like ERA5) may benefit from augmenting them with regional DA using available small- and meso-scale observations, including surface characteristics (e.g., soil properties).

**Code and data availability**

Codes of COSMO model and relevant pre- and post-processing tools are the intellectual property of the COSMO

Consortium and are not publicly available. The Meteosat data are available under the EUMETSAT's Data Policy rules. Initial and boundary conditions are publicly provided by DKRZ (Germany) and contain the Copernicus Climate Change Service Information (ERA-5 reanalysis, 2019) technically processed by the COSMO-CLM Community. The SYNOP and TEMP observations were obtained from the GTS systems of WMO (for Europe) and from the database of Institute of Meteorology and Water Management – National Research Institute (for

Poland). The latter are available via the https://danepubliczne.imgw.pl/ web interface.

**Supplement**

The supplement was submitted with the manuscript.

**Author contributions**

MZZ and WWG conceived the study and advised DKW during its execution. DKW designed, programmed and

ran model simulations, and drafted the initial version of the manuscript. All three authors were involved in drafting its subsequent versions.

**Competing interests**

The authors declare that they have no conflict of interest.

**Acknowledgments**

The paper is based on and extends the results of DKW PhD study at Institute of Meteorology and Water Management – National Research Institute. Numerical experiments were carried out using the COSMO model and

other software maintained by the COSMO Consortium. Computing resources for the experiments were provided by the Institute. External parameters for numerical simulations were obtained by the WebPEP service that is kindly provided by the COSMO-CLM Community. The figures were prepared using NCL software developed at NCAR.

The comments of three anonymous reviewers helped to improve the manuscript.

**Financial support**

This research was supported by Institute of Meteorology and Water Management – National Research Institute in Poland.  NSF National Center for Atmospheric Research is sponsored by the NSF under Cooperative Agreement 1852977.

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
