# Peer review of "Numerical simulation of a rapidly developing bow echo over northeastern Poland on 21 August 2007 using near-grid-scale stochastic convection initiation"

_EGUsphere, 2025_

## Referee Comment (RC1)

**Review comments**

The manuscript titled "Numerical reconstruction of a rapidly developing bow echo over northeastern Poland on 21 August 2007 using near-gridscale stochastic convection initiation" presents a high-resolution numerical simulation of a severe bow echo, using the COSMO model and an ensemble-based stochastic convection initiation scheme. The topic is of clear relevance to the field of mesoscale meteorology, and the study addresses an important and persistent challenge in convective system forecasting.

The paper is well structured, and it clearly outlines both the scientific motivation and the modeling approach. The use of stochastic perturbations to simulate coldpool-driven convection within an ensemble framework is innovative and shows promise in improving the forecast of such simulations. The manuscript also benefits from a comparison with observations, and the additional experiment including sensitivity to wind shear adds useful insight.

However, the manuscript is overly long, in part due to the high number of figures with multiple panels. Some streamlining would enhance the clarity and readability of the paper. Additionally, while the methodology is generally well-described, the manuscript would benefit from a more thorough discussion of the limitations of the experimental setup, particularly with respect to the convection initiation scheme and the ensemble configuration.

In the following, I address specific issues in more detail with a list of comments, together with suggestions for improvements.

**Major Comments**

1. Throughout the manuscript: I recommend not to reference figures in the supplement. The manuscript should contain all the figures that are necessary to show the main results and prove the main points. A sentence can be added in each chapter, informing the readers that more (complete) figures can be seen in the supplement and "not shown" should replace most of the referenced supplement figures. Alternatively, less important statements, especially if they don't bring additional insight, can be completely omitted.
2. L77-85. A sentence or two should be added on these studies' success in alleviating the problem and a better connection to the proposed scheme should be made. How does the proposed scheme address the limitations of existing schemes?
3. L377-380. The paragraph on the shallow convection parameterization can be omitted completely, since its effect is negligible, together with all the related experiments in Table 1 and the related figures and references.
4. Table 2 and 3. The presentation of results would improve if tables 2 and 3 were combined. For example, columns with alternating EX and EM members. The authors could also add text colors or shading of cells based on the values to enhance the differences.
5. L640-641. The ending of this chapter requires more discussion and a better justification for keeping chapter 8. Did the authors see something new, something

expected, something unexpected, in the development of the RIJ? The results presented in this chapter are interesting, but it is unclear which scientific question is being answered.

6. The conclusions would benefit from some more discussion on the limitations of the experimental setup and the CI scheme (e.g. conservation requirements). Moreover, the ensemble properties and its limitations should be discussed (ensemble size, reliability). Considering the high uncertainty of such convective events, is it expected that, even after introducing additional perturbations, all of the ensemble members produce a bow echo?

**Minor Comments**

1. L1. Consider changing the word "reconstruction" with "simulation" in the title and throughout the manuscript.
2. L14-15. Reword, e.g. "The implementation of a new stochastic convection initiation scheme in a 9-member ensemble enables the reconstruction of the event as a cold-pool-driven convective system, with peak gusts closely matching the observed values."
3. L17. Add "wind" to "vertical shear": "vertical **wind** shear".
4. L22. Specify: "delay" with respect to which reference? (reference simulation or observations).
5. L42. Remove "the" in "*the* cold-pool-driven systems".
6. L92. "coincides **with** the scale of contemporary NWP…" – remove "the" (second).
7. L93. Reword "allows…" with e.g. "the scheme facilitates the representation of a high impact…"
8. L104. Why are only surface observations being assimilated? Is it because of a lack of other kinds of observations (e.g. radiosondes)?
9. L108. Add "an" to "After **an** analysis…" or reword "After analysing…".
10. Figure 1. Keep only panels b, d, e, f: the two other pressure levels do not give much more insight, the description in the text is enough. Use different color palettes for panels b, d, as they show different variables and using the same palette can cause confusion. Write the time instead of the pressure level in the top right corner. Increase size of letters M, L indicating the locations. Add latitude and longitude ticks and labels, as the reader might not be familiar with the area of interest.
11. Figure 2. panel b: consider keeping only abbreviations for Mik., Ket. and Leg.
12. Figure 3. Keep panels for 1100, 1130, 1200.
13. Figure 4. I suggest omitting the figure. If the authors want to keep it, they should show only 1 or 2 panels (e.g.1400 or 1500). Keep K, M in all panels for better orientation.
14. L229-232. The paragraph describing the observations in Kaliningrad is not relevant, omit it.
15. Figure 5. Adjust the color scale to make the complete Mediterranean blue.
16. Table 1. Too long and detailed for the experiments. Remove EXSC, EMSC and EX0S. Consider renaming E7-D to E7-CM and EM to EXM (EXM0 to EXM8) for a more systematic structure.

17. Figure 6. I suggest switching E7-A with E2-A for panel a. It would be a more appropriate comparison with the other experiments shown in the figure, as it has the same resolution.
18. L310-314. Are lake effects relevant for the discussion or the experiments in general? I suggest omitting this paragraph or shortening it to a sentence.
19. L330-334. Which corrections were chosen and applied?
20. Figures 7, 10, 17. Increase the size of M (text and dot). In the bottom row of panels, increase contrast of text, e.g. make it white.
21. Figure 8. Omit or move figure to supplement. If kept, a figure showing anomalies with respect to a reference (e.g. the EX experiment) would be more informative.
22. L394-396. Rewrite the information in brackets as a full sentence.
23. Figure 9. Omit figure or move to appendix. The structure of perturbations is clear from the description.
24. L408-409. Why is the 10-minute pause needed?
25. L429-433. This sentence seems out of place or without a clear connection to the previous sentences. A sentence could be added, e.g. "These mechanisms were not studied and are beyond the scope of this paper".
26. Figures 11, 18. Move to supplement. Figure 12 (or 19) is sufficient for the discussion.
27. Figure 12. I suggest choosing 3 members/panels (e.g. best, worst, average). The authors could combine it with figure 19 for a direct comparison (showing the same 3 members). The same applies for figures 13 and 20.
28. Figures 14, 21. Move to supplement. The short discussion does not justify keeping the figure.
29. Figure 15. Consider showing the two fields and trajectories in one panel or keeping the same geographical area in both panels for clarity.
30. L512-513. "horizontally uniform over the area": the figure suggests that this is not true for temperature perturbations. Please clarify.
31. Figure 16. Consider omitting this figure (referenced twice), not essential for the discussion.
32. L531. A figure in the main manuscript clearly showing the effect of modifications on the vertical wind shear amplitude would be insightful (e.g. following the suggestion in the previous comment).
33. L544-547. Omit, see major comment number 3.
34. Figure 17. This figure could show the difference with respect to the EX0 ensemble member.
35. Figure 22. Consider keeping only one panel (e.g. 1500). A possibility would be to add one panel from figure 20 and 21 to show the impact on wind gusts.
36. L647. It is better not to reference figures in the conclusions.
37. L662. "The simulations **respond well**…".

---

## Author Comment (AC1)

Author's response to the reviewers' comments on the manuscript "Numerical reconstruction of a rapidly developing bow echo over northeastern Poland on 21 August 2007 using near-grid-scale stochastic convection initiation."

We would like to thank the Reviewers for their comments and suggestions to improve our manuscript. We considered them carefully and answered every specific comment below. We also extensively amended the text of the manuscript in response to the comments. The Reviewers' comments are in black, our answers are presented in dark orange, while the new proposed text is shown in blue.

At first, however, we need to point that we have found that due to a coding error, the actual temperature perturbations we use are more pronounced than we described in the original manuscript; while the amplitude of temperature perturbation in the central grid column of a perturbation thermal ( $\Delta T$ ) is as described in the original manuscript, the temperature perturbations in the 4 grid columns adjacent to the central perturbation column have the temperature perturbation equal 0.999  $\Delta T$  (which is practically  $\Delta T$ ), and not 0.09  $\Delta T$  as described in the original manuscript. The difference between the perturbations can be characterized by an effective horizontal size of the perturbations. It can be defined as the length of a square which has a uniform temperature perturbation  $\Delta T$  and has the summary temperature excess over its area equal to the summary temperature excess over all grid columns of the perturbation thermal. Actually, the summary temperature excess of the perturbation is practically 5  $\Delta T$  (one grid column with  $\Delta T$  excess and four columns with 0.999  $\Delta T$  excess, each) and not 1.36  $\Delta T$  as assumed in the original manuscript. Therefore, the actual effective horizontal size of the perturbations is 2.2  $\Delta x$  (where  $\Delta x$  is the horizontal grid size of 2.2 km, and the metric size is 4.9 km) and not 1.2  $\Delta x$  (2.6 km) as assumed in our original manuscript. Thus, the effective horizontal size of the perturbation is a near-grid-size value and is still much less than the effective model resolution of 7  $\Delta x$ . With an increase of the effective size of the perturbation, it should be interpreted as representing a flow variability related to the observed 3-to-5-km size convective cells, rather than to 1-3-kmsized large thermals (Marguis et al. 2021). In the revised text, we argue that with that correction, the perturbation amplitudes can no longer be regarded as realistic but are likely amplified by a factor of 2-3 (compared to our assessment of their physical values), which allows to effectively counteract their damping by the model.

We discuss these problems in our answers to the Reviewers' comments (especially while answering Reviewer 3), and the manuscript is accordingly amended and extended, including the Abstract, Introduction, the description of the perturbations (Section 4.2), as well as Summary and Conclusions.

**Reviewers' comments**

**Reviewer 1**

The manuscript titled "Numerical reconstruction of a rapidly developing bow echo over northeastern Poland on 21 August 2007 using near-gridscale stochastic convection initiation" presents a high-resolution numerical simulation of a severe bow echo, using the COSMO model and an ensemble-based stochastic convection initiation scheme. The topic is of clear relevance to the field of mesoscale meteorology, and the study addresses an important and persistent challenge in convective system forecasting.

The paper is well structured, and it clearly outlines both the scientific motivation and the modeling approach. The use of stochastic perturbations to simulate coldpool-driven convection within an ensemble framework is innovative and shows promise in improving the forecast of such simulations. The manuscript also benefits from a comparison with observations, and the additional experiment including sensitivity to wind shear adds useful insight.

However, the manuscript is overly long, in part due to the high number of figures with multiple panels. Some streamlining would enhance the clarity and readability of the paper. Additionally, while the methodology is generally well-described, the manuscript would benefit from a more thorough discussion of the limitations of the experimental setup, particularly with respect to the convection initiation scheme and the ensemble configuration.

Answer: Thank you for the general opinion. The specific comments of the Reviewer are answered below and the manuscript is corrected, accordingly.

In the following, I address specific issues in more detail with a list of comments, together with suggestions for improvements.

**Major Comments**

1. Throughout the manuscript: I recommend not to reference figures in the supplement. The manuscript should contain all the figures that are necessary to show the main results and prove the main points. A sentence can be added in each chapter, informing the readers that more (complete) figures can be seen in the supplement and "not shown" should replace most of the referenced supplement figures. Alternatively, less important statements, especially if they don't bring additional insight, can be completely omitted.

Answer: Following the recommendation, the references to supplement figures are removed and replaced by the phrase "not shown". Also, a sentence is added at the end of Section 1 (Introduction) informing that:

"Supplement contains additional figures supporting the discussion of Sections 4 and 5 (for most of "not shown" remarks)."

2. L77-85. A sentence or two should be added on these studies' success in alleviating the problem and a better connection to the proposed scheme should be made. How does the proposed scheme address the limitations of existing schemes?

Answer: The previous studies on stochastic convection initiation (CI) differ significantly from our approach not only by the applied perturbation method, but also by their scientific goals, as they are mainly interested in a statistically measured overall impact of the schemes on precipitation and some atmospheric-state parameters. Following the comment, the discussion on previous studies and their results was extended, showing their limited statistical performance and forming a better context for the proposed scheme. Our work has a different and more basic goal as it aims at a realistic representation of a specific severe convective development. Thus, our CI scheme needs to ingest sufficient amount of small-scale flow variability, which would allow the model to engage with the imposed perturbations and to develop a realistic, organized convective system. To achieve our goal, we use perturbations that stir the flow on a near-grid scale, which coincides with the scale of observed initial convective cells (Marguis et al. 2021). The drawback of our method is that it requires an overestimation of the perturbation's amplitude (likely by a factor of 2-3) to counteract nonphysical damping of the perturbations by the model. Two paragraphs from former lines 77-96 are extended and replaced by the following text in current lines 83-117.

[revised manuscript text omitted]

3. L377-380. The paragraph on the shallow convection parameterization can be omitted completely, since its effect is negligible, together with all the related experiments in Table 1 and the related figures and references.

Answer: We agree with the Reviewer's requirement to make the manuscript possibly concise, including the description of the application of shallow convection parameterization. Considering also the minor comment 5 of Reviewer 2, who requires an extension of the shallow convection parameterization experiment with an additional experiment on the

sensitivity of CI to a tuning of the turbulence parameterization, we propose to possibly shorten the relevant discussion on the issue and complement it with a brief comment on the related experiment required by Reviewer 2, together with removing the convection parameterization experiments from Table 1, removing the former Figure 8 and removing the information on shallow convection experiment from former Section 7.2 (current Section 5.2). A sentence in the current Section 7 summarizes the results of both experiments. The modified text, responding to both Reviewers' comments, is as follows (lines 359-364):

"Additional measures like the application of shallow convection parameterization (recommended by Doms et al., 2021) did not improve the forecast (convective gusts below 20 m s-1 by 15:00). Also, the reduction of asymptotic maximum turbulence mixing length scale tur\_len of turbulence parameterization from recommended 150 m (used in our experiments) to 75 m (which is known to help in CI on cost of low-tropospheric warm temperature bias; Baldauf et al., 2011) marginally improves the forecast with maximum gusts reaching only 19 m s-1 until 14:30 and locally 23 m s-1by 15:00 (not shown)."

And in "Summary and conclusions", the modified sentence is (current lines 647-650): "Indeed, we demonstrate that the operational convective-scale COSMO model with a horizontal grid spacing of 2.2 km used for our study encounters significant problems with numerical reconstruction of the event, even with favorable atmospheric environmental conditions and additional application of shallow convection parameterization or reduced maximum turbulence mixing length tur\_len."

4. Table 2 and 3. The presentation of results would improve if tables 2 and 3 were combined. For example, columns with alternating EX and EM members. The authors could also add text colors or shading of cells based on the values to enhance the differences.

Answer: Following the Reviewer comment, Tables 2 and 3 are combined to a single Table 2 with EX- and EM-ensemble members distinguished by blue/red colors.

5. L640-641. The ending of this chapter requires more discussion and a better justification for keeping chapter 8. Did the authors see something new, something expected, something unexpected, in the development of the RIJ? The results presented in this chapter are interesting, but it is unclear which scientific question is being answered.

Answer: The discussion was extended, as requested, following also major comment 5 of Reviewer 2. We point that the model's problems with the development of a bow-like cloud structure (due to the model limitations discussed in the Introduction) indicate that the model may also struggle with the representation of other physical aspects of the analyzed convective system, including the RIJ formation. Current Section 6 demonstrates that, fortunately, it is not the case. The first paragraph of the new Section 6 was extended via the following text (current lines 596-602):

"The model's problems with the development of a bow-like cloud structure (related to the model limitations discussed in the Introduction) indicate that the model may also struggle with representation of other physical aspects of the analyzed convective development,

including the RIJ formation. Therefore, a brief analysis of how the model-represented convective processes influence the low-to-mid tropospheric flow is performed. The EM0 forecast is chosen for the purpose because it gives the timing and position of the convective system most closely resembling the observations, and wind gusts in the period between 13:00 and 14:00 are the strongest (38 m s-1). It may be assumed that the forecast gives the best approximation to the real development."

Additionally, the last paragraph of the new Section 6 was extended as follows (current lines 633-638):

"The analysis shows a lack of significant model problems in the RIJ simulation: the strong RIJ forms already at the very early stage of the convective system development, that is at 13:00. Moreover, the resulting convective circulation allows the RIJ to influence the magnitude of the 10-m wind and gusts already at that early development stage (and later). The model limitations do not significantly impair its ability to simulate the RIJ formation. The simulation strongly suggests that an early developing RIJ may notably influence the catastrophic strength of convective gusts actually observed at the early stage of the developing bow echo near 13:00."

6. The conclusions would benefit from some more discussion on the limitations of the experimental setup and the CI scheme (e.g. conservation requirements). Moreover, the ensemble properties and its limitations should be discussed (ensemble size, reliability). Considering the high uncertainty of such convective events, is it expected that, even after introducing additional perturbations, all of the ensemble members produce a bow echo?

Answer: Following the comment, as well as the requirements of Reviewer 3, the discussion on the limitations of our experimental setup was significantly extended both within the description of the CI scheme (Section 4.2) and "Summary and conclusions" Section of the manuscript. The extended discussion concerns also the ensemble size and reliability. Answering the reviewer's question, we expect (based on the arguments presented below in the new manuscript text) that with the introduction of additional ensemble members (driven by the stochastically produced CI perturbations) the ensemble would stay reliable with the majority of its members forecasting the development of a cold-pool-driven convective system (because of a speculative character of the sentence we propose not to include it in the text of the manuscript). The proposed new text also responds to minor comment 6 of Reviewer 2. The following paragraphs were added to Section 4.2 (lines 374-421):

[revised manuscript text omitted]

**Minor Comments**

1. L1. Consider changing the word "reconstruction" with "simulation" in the title and throughout the manuscript.

Answer: We use the word "reconstruction" also in its wider sense of "an attempt to get a complete description of an event using the information available" (Cambridge Dictionary), also to point out that our work goes beyond a straightforward or relatively simple simulation. One of the reasons is that we are working on the margin of applicability of NWP models for a representation of convective processes (as we discussed in the Introduction). Following the comment, we replaced the word "reconstruction" with "simulation" where its meaning was close to the latter, including the title (lines 1, 13, 247, 640), and left it unchanged where its meaning was more general, as in the definition cited above (lines 15, 19, 90, 145, 249, 291, 648).

2. L14-15. Reword, e.g. "The implementation of a new stochastic convection initiation scheme in a 9-member ensemble enables the reconstruction of the event as a cold-pool-driven convective system, with peak gusts closely matching the observed values."

Answer: The request is implemented.

3. L17. Add "wind" to "vertical shear": "vertical wind shear".

Answer: The request is implemented.

4. L22. Specify: "delay" with respect to which reference? (reference simulation or observations).

Answer: The manuscript is modified accordingly (delay with respect to observations).

5. L42. Remove "the" in "the cold-pool-driven systems".

Answer: The request is implemented.

6. L92. "coincides with the scale of contemporary NWP..." – remove "the" (second).

Answer: The request is implemented in the text of the modified sentence.

7. L93. Reword "allows..." with e.g. "the scheme facilitates the representation of a high impact..."

Answer: The request is implemented in the text of the modified paragraph.

8. L104. Why are only surface observations being assimilated? Is it because of a lack of other kinds of observations (e.g. radiosondes)?

Answer: The reliability of ERA5 upper air assimilation system results from balanced assimilation of different types of upper air data, including not only radiosondes (which are sparse in space and time) but also (much more dense) satellite measurements. The mesoscale environment in the vicinity of the convective area is characterized by strong spatial variability (Figs. 1 and 2). In such circumstances our additional assimilation of the sparse radiosonde data may likely lead to a destruction of that balance and a significant deterioration of upper-air analysis, especially if the radiosondes' measurements are not representative for the convection area. This is the case of Legionowo measurements at 12 UTC (in terms of wind, CAPE, CIN; see section 2), even though the station is located close to the convective area.

Following the comment, the following sentence is added to the text of the manuscript (current lines 132-134):

"The aerological soundings are not additionally assimilated mainly due to problems with their representativity for the environment of the developing convection (see Section 2.2)."

9. L108. Add "an" to "After an analysis..." or reword "After analysing...".

Answer: The request is implemented: "After analysing ..." is used.

10. Figure 1. Keep only panels b, d, e, f: the two other pressure levels do not give much more insight, the description in the text is enough. Use different color palettes for panels b, d, as they show different variables and using the same palette can cause confusion. Write the time instead of the pressure level in the top right corner. Increase size of letters M, L indicating the locations. Add latitude and longitude ticks and labels, as the reader might not be familiar with the area of interest.

Answer: The request is implemented. Additionally, following the comment of Reviewer 3, the figure was split into new Fig. 1 showing a large European domain, and Fig. 2 for smaller domain.

11. Figure 2. panel b: consider keeping only abbreviations for Mik., Ket. and Leg.

Answer: We propose to keep the abbreviations for the station names. It allows the reader to identify the locations of weather stations providing data for our modifications of the PBL characteristics. The stations are listed in the discussion in Sections 3.2.1 and 3.2.2. However, the abbreviations for Mik., Ket. and Leg. are written in bold to distinguish these stations. This is currently Fig. 3.

12. Figure 3. Keep panels for 1100, 1130, 1200.

Answer: A new version of the Figure fulfilling the reviewer's requirement was prepared and implemented; it is now Fig. 4.

13. Figure 4. I suggest omitting the figure. If the authors want to keep it, they should show only 1 or 2 panels (e.g.1400 or 1500). Keep K, M in all panels for better orientation.

Answer: The figure is of special value for our analysis as it serves as the observational benchmark for the model representations of the bow-like structure of the convective system, together with its position and timing. We propose, therefore, to keep the figure with all its panels. However, its modified version with letters K, M in all panels was prepared. It is now Fig. 5.

14. L229-232. The paragraph describing the observations in Kaliningrad is not relevant, omit it.

Answer: The request is implemented and the paragraph is removed.

15. Figure 5. Adjust the color scale to make the complete Mediterranean blue.

Answer: The figure was corrected accordingly; it is now Fig. 6.

16. Table 1. Too long and detailed for the experiments. Remove EXSC, EMSC and EX0S. Consider renaming E7-D to E7-CM and EM to EXM (EXM0 to EXM8) for a more systematic structure.

Answer: The experiments EXSC, EMSC, and EX0S are removed from Table 1. We would prefer, however, to stay with the names of the EM and EM-ensemble without changing them. The current names are possibly short and well distinguish the ensembles. There are also many figures, especially in the Supplement, with the names of ensemble members printed on them. The renaming would require substantial additional work. Following the intention of the Reviewer's proposition, we changed the name of E7-D to E7-M for a more systematic structure.

17. Figure 6. I suggest switching E7-A with E2-A for panel a. It would be a more appropriate comparison with the other experiments shown in the figure, as it has the same resolution.

Answer: The former Fig. 6 is removed following the further-reaching major comment 2 of Reviewer 2.

18. L310-314. Are lake effects relevant for the discussion or the experiments in general? I suggest omitting this paragraph or shortening it to a sentence.

Answer: The reviewer is right. Following the comment, we removed the paragraph on lake effects.

19. L330-334. Which corrections were chosen and applied?

Answer: The description of soil moisture correction was modified and shortened appropriately, following also major comment 2 of Reviewer 2, with the following new text (current lines 326-331):

"Since the soil moisture measurements are not available, several plausible alternatives were tested to minimize the simulated 2-m T and Td biases across 13 WSs in northeastern Poland (Białystok, Elbląg, Kętrzyn, Mikołajki, Mława, Olsztyn, Ostrołęka, Siedlce, Suwałki, Terespol) and western Belarus (Baranovichy, Grodno, Lida; see Wójcik, 2021). The finally implemented relative soil moisture corrections vary between 50% (e.g., Olsztyn, Terespol) and -50% (e.g., Kętrzyn, Ostrołęka), see column 5 of table 5.2 in Wójcik (2021)."

20. Figures 7, 10, 17. Increase the size of M (text and dot). In the bottom row of panels, increase contrast of text, e.g. make it white.

Answer: Following the request, the size of M and the relevant dot was increased in new Figures 8 and 13, equivalent to former Figures 10 and 17. The former Figure 7, showing the evolution of environmental conditions for experiment EX, was removed following the further-reaching major comment 3 of Reviewer 2. Because of a change in the color scale in the bottom row of panels, resulting from a minor comment 7 of Reviewer 2, there was no need to change the text and dot color there.

21. Figure 8. Omit or move figure to supplement. If kept, a figure showing anomalies with respect to a reference (e.g. the EX experiment) would be more informative.

Answer: The former Figure 8, showing maximum gusts of experiments without successful CI, is removed from the manuscript, following also the major comment 3 of Reviewer 2. The modified figure, including also maximum gusts from the additional experiment with decreased maximum turbulence mixing length scale (required by minor comment 5 of Reviewer 2), is moved to the Supplement. We propose that this Supplement figure directly

shows maximum gusts values rather than their anomalies because the former offers direct information on the strength of the simulated convection and is directly comparable with available observations.

22. L394-396. Rewrite the information in brackets as a full sentence.

Answer: The description of the perturbation is changed and the old description is removed.

23. Figure 9. Omit figure or move to appendix. The structure of perturbations is clear from the description.

Answer: The former Figure 9 is removed, especially that it was not correct.

24. L408-409. Why is the 10-minute pause needed?

Answer: The idea was to allow the perturbations of an active period to grow further undisturbed by the next perturbations, during the pause. However, we did not test whether that pause is of practical value in the process. The issue is concisely addressed in Section 7 while also answering major comment 6 of Reviewer 1, minor comment 6 of Reviewer 2 and the comment of Reviewer 3. The following text was added in lines 666-672:

"As for the limitations of our experiment setup, we already discussed its temperature bias and perturbation amplitudes likely stronger (2-3 times) than realistic. We also did not optimize the method in terms of spatial and temporal density, nor the final amplitudes and shapes of the perturbations. We think that the proposed CI scheme may be useful for deep convection studies. Such a perturbation strategy, corrected for its temperature bias, may still be of interest for the NWP applications because it allows continuous stirring of near-grid-scale variability in a way that accounts for the model's dissipative properties. We consider it reasonable that future applications of such a CI procedure will deliberately use near-grid-scale but overestimated temperature perturbations."

25. L429-433. This sentence seems out of place or without a clear connection to the previous sentences. A sentence could be added, e.g. "These mechanisms were not studied and are beyond the scope of this paper".

Answer: We added the sentence, as recommended.

26. Figures 11, 18. Move to supplement. Figure 12 (or 19) is sufficient for the discussion.

Answer: The request is implemented and the former Figures 11 and 18 are moved to the Supplement.

27. Figure 12. I suggest choosing 3 members/panels (e.g. best, worst, average). The authors could combine it with figure 19 for a direct comparison (showing the same 3 members). The same applies for figures 13 and 20.

Answer: Thank you for the suggestion, it is implemented as follows. The best/average/worst ensemble members are defined based on the strength of their maximum gusts from Table 2. The ensemble members EX0/EX2/EX3 (respectively) are chosen as representatives of these solution classes. As suggested, they are shown together with EM0/EM2/EM3 members for an analysis of pseudo-reflectivity and convective updrafts' structure in EX- and EM-ensembles. New Figures 9 (for pseudo-reflectivity) and 10 (for convective updrafts structure) for ensemble members EX0/EX2/EX3 and EM0/EM2/EM3 at 14:00 are implemented into the manuscript. Former Figures presenting all ensemble members are moved to the Supplement.

28. Figures 14, 21. Move to supplement. The short discussion does not justify keeping the figure.

Answer: Former Figures 14 and 21 show the forecasted spatial distribution and strength of maximum gusts in comparison to available proxy information on damages and fatalities. The maximum gusts are the main indicator of the convective system's strength and its potential social impact. Therefore, the Figures inform on practically and socially important aspects of the simulations. We propose, therefore, to keep the information -albeit in a reduced form- in the main manuscript. We propose, in the spirit of the previous comment of the Reviewer, to reduce former Figures 14 and 21 to a single Figure 11 showing the information for best/average/worst ensemble members of both EX- and EM-ensembles (in analogy to new Figures 9 and 10). Former Figures 14 and 21, showing the information for all ensembles' members, are moved to the Supplement.

29. Figure 15. Consider showing the two fields and trajectories in one panel or keeping the same geographical area in both panels for clarity.

Answer: The suggestion is implemented, and a new Figure 12 showing both perturbation fields and all trajectories in one panel is implemented into the manuscript.

30. L512-513. "horizontally uniform over the area": the figure suggests that this is not true for temperature perturbations. Please clarify.

Answer: The former Figure 15 (and current 12) is correct, as the temperature itself is modified to become uniform in the perturbation area, but to achieve this effect, the temperature perturbation varies according to the spatial variability of the non-perturbed temperature before modification. The modified text stresses that it is the finally modified temperature field that is constant in the source area (lines 520-525):

"The temperature modifications for the eastern side of the increased temperature gradient are performed in the source area of trajectories arriving east of Mikołajki at 12:00, located near and south of the Shepetivka upper-air station (Fig. 11 for the applied temperature perturbations). The modification area is contained between 900 and 520 hPa and has the form of an ellipse having foci located at 27.05°E, 50.18°N, and 28.10°E, 47.25°N; the semi-major axis equals 201 km. The temperature is increased to measurements from Shepetivka sounding at 00:00 and is modified to be horizontally uniform over the area."

31. Figure 16. Consider omitting this figure (referenced twice), not essential for the discussion.

Answer: The suggestion is implemented and former Figure 16 is removed.

32. L531. A figure in the main manuscript clearly showing the effect of modifications on the vertical wind shear amplitude would be insightful (e.g. following the suggestion in the previous comment).

Answer: New Figure 7 presenting EX shear, EM shear and their difference at 12:00 was prepared and is implemented in the manuscript.

33. L544-547. Omit, see major comment number 3.

Answer: The request is implemented, and the text concerning shallow convection parameterization is removed here.

34. Figure 17. This figure could show the difference with respect to the EX0 ensemble member.

Answer: Following the comment, a new Figure presenting perturbations relative to the EX0 forecast was prepared and is presented below. The Figure is, however, rather complicated and would require a relatively extended explanation and discussion, especially for 12:00 and 13:00, when the perturbations resulting from environmental changes between EX and EM settings, CI perturbations developing in different environments, and the developing deep convection, interact also with themselves. Therefore, in the interest of keeping the text possibly concise, we propose to modify the former Figure 17 according to minor comment 20 of the Reviewer and include it in the manuscript as new Figure 13, because it is much easier for a concise interpretation.

Figure 13: The environmental conditions in the vicinity of northeastern Poland on 21 August 2007: differences between EM0 and EX0 forecast: MCAPE difference (J kg-1, top row), 100 to 3000 m vertical shear difference (m s-1, second row), MCIN difference (J kg-1, third row, note different CIN scales below and above 10 J kg-1), all at 10:00 (left column), 12:00 (middle column) and 13:00 (right column); black contour shows pseudo-reflectivity of 30 dBZ at altitude of 3000 m in EM0 forecast, black dots show the positions of Mikołajki (M) and Legionowo (L).

35. Figure 22. Consider keeping only one panel (e.g. 1500). A possibility would be to add one panel from figure 20 and 21 to show the impact on wind gusts.

Answer: Former Figure 22 not only shows the horizontal distribution of wind speed but also allows us to indicate where the cross-sections shown in former Figure 23 are located relative to the structure of the convective system. We propose, therefore, to keep the former Figure 22 in the main text of the manuscript as current Figure 14.

36. L647. It is better not to reference figures in the conclusions.

Answer: Following the recommendation, the comment on the convergence zone is moved to Section 5.1 and only referenced in the "Summary and conclusions". The text in Section 5.1, in the current lines 512-516 is modified as follows:

"The experiment is set up via appropriate modifications of lower-to-mid-tropospheric temperatures in the source area of air that departs at 00:00 and reaches the colder/warmer (western/eastern) side of the high-wind band over Mikołajki around 12:00. The source areas are defined using trajectory analysis (Fig. 11; the trajectories are calculated with Lagranto software following Sprenger and Wernli, 2015), which also shows that the band coincides with a local convergence zone indicated by the converging trajectories."

The text in Section 7, in current lines 641-643 is modified as follows:

"The system was weakly forced in the sense of the lack of the omega equation forcing for strong synoptic-scale ascent or nearby frontal surfaces. However, it developed in the vicinity of a local convergence zone (Section 5.1)."

37. L662. "The simulations respond well...".

Answer: The correction is implemented.

**Reviewer 2**

The paper presents a report on numerical experiments simulating a bow echo that impacted Poland in 2007. It shows that introducing small-scale temperature perturbations into the ERA5 analysis enables the COSMO model to simulate the bow echo correctly. The study is interesting, as the authors successfully simulate the bow echo and discuss the roles of cold pools and vertical wind shear. However, the paper is excessively long. Many sections contain unnecessary details that can be omitted, while other parts require more thorough discussions. I recommend a major revision.

Answer: Thank you for the general opinion. The specific comments of the Reviewer are answered below and the manuscript is corrected, accordingly.

**Major comments**

1. Section 2, which describes the bow echo, could be shortened.

Answer: Section 2 is shortened: its introductory part is removed, and Subsections 2.2 and 2.4 are reduced. In the latter, a discussion on convective system observations in Kaliningrad is removed. Also, the content of Fig. 3 is reduced. The changes also follow minor comments 12 and 14 of Reviewer 1.

2. Section 4, which "discusses a reconstruction of the initial conditions for prognostic experiments", could be merged with Section 3. This is because the changes in soil and surface conditions are kept in the subsequent experiments with CI. Furthermore, the results from nudging of soil and surface observations, as well as the modifications of surface heat fluxes, are not summarized in either the conclusion or the abstract. Therefore, the associated figures can be omitted while a concise description of the soil and surface changes, along with a brief summary, could be retained.

Answer: Following the request, former Section 4 is merged with Section 3 as Subsection 3.2, and the former Fig. 6 is removed. The descriptions of soil and surface temperature changes are shortened. The relevant text of the manuscript was modified as follows:

**Subsection 3.2.1: "Nudging of soil and surface observations", current lines 306-318:**

"We improve these 2-m T and Td biases in a few steps. First, nudging of routine soil temperature measurements from 12 WSs in northeastern and eastern Poland (Mikołajki, Siedlce, Olsztyn, Białystok, Terespol, Mława, Elbląg, Suwałki, Warszawa-Okęcie, Kozienice, Włodawa, and Lublin) is performed using the simulation E7-B. It starts the previous day (20 August) at 00:00 using C-7 with IC and BC from ERA5. Simulation E7-B provides the IC for soil temperature at 00:00 of 21 August for the corrected C-7 simulation starting at that time (E7-C). The nudging continues within the main C-2 simulation (experiment E2-B) starting at 00:00 on 21 August and lasts until 13:00 with the soil IC taken from E7-B. The atmospheric IC and BC are downscaled from ERA5 using E7-C. The experiment E2-B additionally performs COSMO nudging of SYNOP observations of 2-m Td, 10-m wind, and surface pressure between 00:00 and 08:00. All available observations within the model domain (Fig. 5) are assimilated including hourly observations from Poland (77 stations) and 3-hourly observations from abroad (225 stations). However, E2-B still incompletely removes the pre-convective 2-m T and Td errors: RMSE for Mikołajki are 1.61°C and 1.07°C, and for Kętrzyn 1.88°C and 1.24°C, respectively."

**Subsection 3.2.2: "Modification of surface heat fluxes", current lines 320-335:**

"As E2-B develops excessive morning cloud cover (compared to satellite observations, not shown), a subsequent experiment EX additionally increases the insolation over northeastern Poland and western Belarus to realistic values characteristic of the cloudless sky, following studies using modified cloud-radiation interactions (e.g., Wu et al., 1998; Harrop et al., 2024 and references therein). The modification is active from 03:30 (approximate sunrise) until 12:00, but is locally turned off if precipitation is detected. The partitioning of the resulting surface heat flux into its sensible and latent components is corrected by altering the initial

moisture content of the topmost 0.2 m deep soil layer following, e.g., Yamada (2008) or Gerken et al. (2015). Since the soil moisture measurements are not available, several plausible alternatives were tested to minimize the simulated 2-m T and Td biases across 13 WSs in northeastern Poland (Białystok, Elbląg, Kętrzyn, Mikołajki, Mława, Olsztyn, Ostrołęka, Siedlce, Suwałki, Terespol) and western Belarus (Baranovichy, Grodno, Lida; see Wójcik, 2021). The finally implemented relative soil moisture corrections vary between 50% (e.g., Olsztyn, Terespol) and -50% (e.g., Kętrzyn, Ostrołęka), see column 5 of table 5.2 in Wójcik (2021).

The applied corrections significantly improve the 2-m T and to a smaller degree  $T_d$  forecasts in the pre-convective period over northeastern Poland. RMSE for Mikołajki is reduced to  $0.61^{\circ}$ C for 2-m T and is  $0.73^{\circ}$ C for  $T_d$ , and for Kętrzyn they become  $1.32^{\circ}$ C and  $1.01^{\circ}$ C, respectively. That brings the CAPE and CIN of EX close to values estimated from available observations, as discussed in the following section."

3. Section 5, which "discusses the results of the experiment without the CI scheme", could be removed. The main relevance of this experiment lies in its failure to reproduce the bow echo. While this provides justification for doing experiments with the CI scheme, the key message could be conveyed in a single sentence. Given that the paper focuses on the CI scheme, it is unclear why so much detail is dedicated to an experiment that does not use it.

Answer: The main purpose of former Section 5 was 4-fold: 1: to demonstrate that after modification of boundary layer conditions, the resulting MCAPE and MCIN agree with observation-based estimations from Section 2.2; 2: to provide general quantitative information on model-derived vertical shear in the convective area; 3: to demonstrate that despite these realistic MCAPE, MCIN and favorable shear the severe convective system was not forecasted by the model; 4: indicate that model-represented development of deep convection has a signature of Peters et al. (2022) dynamics (we consider a corroboration of their finding an important part of our study). We propose keeping this information in the manuscript, considering it important for our discussion.

Following the Reviewer's comment, the former Section 5 was removed together with its Figures 7 and 8. However, we propose that a possibly brief summary concluding the above points 1 to 4 serves as an introduction to Section 4, "Convection initiation scheme and its impact" (as Section 4.1). On request of Reviewer 1 (point 32 of minor comments), new Fig. 7 presents vertical shear at 12:00 from EX- and EM-forecasts together with their difference. The relevant text of the manuscript, which also accommodates minor comment 5 of the Reviewer, was modified as follows:

Subsection 4.1: "Convection forecast without convection initiation scheme", current lines 338-364:

"The C-2 simulation with the corrected CBL characteristics, referred to as the EX-forecast, does not develop severe convection over northeastern Poland, despite reproducing the atmospheric environmental conditions in agreement with observation-based estimations from Section 2.2. Local maxima of MCAPE reach locally 2900 and 3300 J kg-1 south of Mikołajki

at 11:30 and 13:30, respectively, while MCIN immediately south-west of Mikołajki attains values below 10 J kg-1 and close to 1 J kg-1 already from 11:00 (not shown). The simulation also shows a band of increased low-tropospheric vertical shear (defined as the difference between the wind vectors at 3000 and 100 m AGL) located southwest of Mikołajki (Fig. 6a). Between 12:00 and 14:00, the area of prominent shear of at least 14 m s-1 within the band slowly moves toward Mikołajki.

The deep convection (defined here as a presence of at least 30 dBZ pseudo-reflectivity at 3000 m AGL) development is noteworthy. It is late, at 12:30, and not in the highest CAPE and low CIN area, but where a belt of increased vertical shear exceeding 15 m s-1 coincides with a belt of locally increased MCAPE exceeding 2300 J kg-1, about 80 km west of Mikołajki (not shown). That strongly suggests deep convection initiation dynamics according to Peters et al. (2022a; b). They showed that, despite earlier considerations, the high shear environment may promote the process via high-shear-induced dynamic pressure perturbations adjacent to sufficiently developed thermals.

However, the convective gusts (calculated following Brasseur, 2001) do not exceed 15 m s-1 until 14:30 and reach only 20 m s-1 by 15:00. With the additional lack of bow-shaped convection organization (not shown), the simulation is not successful. Additional measures like the application of shallow convection parameterization (recommended by Doms et al., 2021) did not improve the forecast (convective gusts below 20 m s-1 by 15:00). Also, the reduction of asymptotic maximum turbulence mixing length scale tur\_len of turbulence parameterization from recommended 150 m (used in our experiments) to 75 m (which is known to help in CI on cost of low-tropospheric warm temperature bias; Baldauf et al., 2011) marginally improves the forecast with maximum gusts reaching only 19 m s-1 until 14:30 and locally 23 m s-1 by 15:00 (not shown)."

4. Section 7.2 on "Impact of shear modification on environmental conditions and deep convection without CI", raises a similar concern. Why is emphasis placed on experiments that do not involve the CI scheme?

Answer: The main purposes of former Section 7.2 were similar to those of Section 5: to demonstrate that MCAPE and MCIN are still in agreement with their observation-based estimations, the vertical shear increased, there was no severe convection development, and the signature of Peters et al. dynamics is still present.

Following the Reviewer's comment, the former Section 7.2 was removed. We propose, however, that the following brief information be retained in the manuscript in the following way. A summary information on how the low-to-mid tropospheric temperature modifications changed the vertical wind shear (without destroying MCAPE and MCIN), with a reference to new Figure 7 required by comment 32 of Reviewer 1, would conclude Section 5.1 "Shear modification technique". Brief summary information that increased shear alone does not give a better convection forecast, and that deep convection development has a signature of Peters et al. (2022) dynamics would form an initial part

of revised Section 5.2 "Convection forecast for increased vertical shear". The relevant text of the manuscript was modified as follows:

In Subsection 5.1: "Shear modification technique", current lines 541-546:

"As expected, the modifications mainly influence the vertical shear amplitude in the EM-forecast while preserving its overall mesoscale pattern over northeastern Poland between 10:00 and 14:00. In the EM-forecast, as in the basic EX-forecast, the shear slowly weakens with time but remains stronger by 2-3 m s-1 in most of the area near Mikołajki, locally even by about 5 m s-1 by 12:00 and by about 7 m s-1 at 13:00, compared to the EX-forecast (Fig. 6bc for 12:00). The thermodynamic conditions are also alike, including MCIN and MCAPE overall patterns (not shown)."

In Subsection 5.2: "Convective forecasts with increased vertical shear", current lines 548-558:

"Overall, the increase in shear alone did not improve the severe weather forecast. In the EMforecast without stochastic CI, deep convection develops at 13:00 about 40 km south of Mikołajki, in the area where a narrow belt of shear reaching 17 m s-1 catches up with a narrow belt of MCAPE exceeding 3000 J kg-1 (not shown), in agreement with the dynamics discussed by Peters et al. (2022a; b). Compared to the EX-forecast, the maximum convective gusts are weaker (down to about 15 m s-1 by 15:00) despite convection developing within the area with accessible stronger CAPE and shear.

Further implementation of the CI scheme within the EM-forecast produces a new EM-ensemble and decisively changes the convection forecast. The ensemble members EM0 to EM8 are generated using the saved configurations of the stochastic CI used to generate ensemble members EX0 to EX8."

5. Section 8 on "Rear inflow jet in the EM0 forecast", includes two figures and half a page of text. This section should either be expanded with a more thorough discussion or removed altogether.

Answer: The discussion in Former Section 8 (current Section 6) is extended following also major comment 5 of Reviewer 1. The relevant modifications to the manuscript are shown above while answering major comment 5 of Reviewer 1.

**Minor comments**

1. Line 26, bow echoes are indeed a specific class of deep moist convection organization. As written Line 60, they are "developing under a significant external forcing". With this respect, how bow echoes can be self organized?

Answer: The full original sentence containing the phrase from line 60 reads:

"It is thus no surprise that the successful bow echo simulations concern mainly systems prone to increased predictability: relatively large or long-lasting (6 or more hours, including

derechos), embedded within large convective systems, or developing under a significant external forcing (e.g., by fronts, Lawson and Gallus, 2016)."

The sentence concerns successful numerical simulations of bow echoes and bow echoes of increased predictability, rather than bow echoes in general. Not all bow echoes develop under significant external forcing. A prominent example is the bow echo analyzed in our study. We argue in Section 2.1 that this severe convective system did not develop "under significant external forcing". To better stress that the discussed sentence concerns numerical simulations of bow echoes and not bow echoes in general, we propose to use the phrase "numerical simulations" instead of "simulations" in our text.

As for the mechanism of bow echo self-organization, it was described in the original version of the manuscript as follows:

"A theory by Rotunno et al. (1988) (known as the RKW theory, see also a discussion in Bryan et al., 2006) explains the systems' organization and persistence via the approximate balance of the horizontal vorticity of the environmental flow and of the sufficiently strong cold pool flow at the pool's leading edge. The balance forces deep lifting and a formation of new convective cells at the leading edge, making bow echoes cold-pool-driven (Coniglio et al., 2005) systems."

Following the request of Reviewer 3, the text was extended for a discussion of the RKW theory. The revised text is as follows (current lines 39-47):

"A theory by Rotunno et al. (1988) (known as the RKW theory) explains the systems' organization and persistence via the approximate balance of horizontal vorticity of the environmental flow and of the sufficiently strong cold pool flow at the pool's leading edge. The balance forces deep lifting and a formation of new convective cells at the leading edge, making bow echoes cold-pool-driven (Coniglio et al., 2005) systems. Further research confirmed the validity of the RKW theory for idealized systems (see Bryan et al., 2006). The studies of real events also confirm the presence of the RKW mechanism for strong low-level shears, while indicating that also the presence of a notable shear above may lead to the development of such severe and persistent systems (Stensrud et al., 2005; Weisman and Rotunno, 2005; Cognilio et al., 2012; Kirshbaum et al., 2025)."

**Additional references:**

Stensrud, D. J., Coniglio, M. C., Davies-Jones, R. P., and Evans J. S.: Comments on "A theory of strong long-lived squall-lines revisited", J. Atmos. Sci., 62, 2989–2996, doi:10.1175/JAS3514.1, 2005.

Weisman, M. L, and Rotunno, R.: Reply, J. Atmos. Sci., 62, 2997–3002, doi:10.1175/JAS3515.1, 2005.

Coniglio, M. C., Corfidi, S. F., and Kain , J. S.: Views on applying RKW theory: an illustration using the 8 May 2009 derecho-producing convective system, Mon. Wea. Rev., 140, 1023–1043, doi:10.1175/MWR-D-11-00026.1, 2012.

Kirshbaum, D J., Sindhu, K. D., and Turner, D. D.: An observational evaluation of RKW theory over the U.S. Southern Great Planes, J. Atmos. Sci., 82, 1341–1360, doi:10.1175/JAS-D-24-0185.1, 2025.

2. Line 17, MCAPE and MCIN are usually named CAPE and CIN.

Answer: The text of Abstract is modified, accordingly.

3. Lines 30 and 52, "Europe [...] and Poland" This wording suggests that Poland is not part of Europe.

Answer: The text is modified, accordingly, and the phrase "including Poland" is used, instead of "and Poland".

4. Line 67, Uncertainties in the representation of turbulence also affect cloud organization (Machado and Chaboureau 2015, Tompkins and Semie 2017)

Answer: The description of the perturbation technique was significantly extended with the corrected characteristics of the perturbations, following also the comments of Reviewer 3. The general issue of the sensitivity of the CI scheme to the values of its different parameters, including the perturbations' shape, probabilistic distribution of their amplitude, as well as their spatial and temporal variability, needs a thorough study. Moreover, such a study should be performed in the wider context of a parallel optimization of the setup of already existing physical parameterizations influencing the CI. A prominent example is the turbulence parameterization, which -as rightly pointed out by the Reviewer in the previous commentinfluences the process, e.g., via a choice of the maximum turbulence mixing length scale tur len. In a result, also the tur len value should be optimized to be based possibly on the physical properties of subgrid turbulence rather than on the parameter's surrogate properties for CI. Another such issue, pointed by Reviewer 3, is the influence of the statistical cloud scheme on the representation of CI. Such analysis lies outside the scope of our paper. In our opinion, the proposed CI scheme can be used in other case studies, especially those experiencing CI problems, possibly after correcting it for the temperature bias implemented by the scheme (see the discussion with Reviewer 3). As for the operational use of the method, Reviewer 3 has a strongly negative opinion on the issue, as the perturbation amplitudes are likely stronger than realistic, to compensate for the unphysical diffusive properties of the model. In our opinion, the method may still be considered for operational use in NWP, after correction for its bias.

To respond to the comment, as well as major comment 6 of Reviewer 1 and comments of Reviewer 3, the following text was added to Sections 1, 4.2 and 7 of the manuscript (lines 666-672):

"As for the limitations of our experiment setup, we already discussed its temperature bias and perturbation amplitudes likely stronger (2-3 times) than realistic. We also did not optimize the method in terms of spatial and temporal density, nor the final amplitudes and shapes of the perturbations. We think that the proposed CI scheme may be useful for deep convection

studies. Such a perturbation strategy, corrected for its temperature bias, may still be of interest for the NWP applications because it allows continuous stirring of near-grid-scale variability in a way that accounts for the model's dissipative properties. We consider it reasonable that future applications of such a CI procedure will deliberately use near-grid-scale but overestimated temperature perturbations."

7. Line 422, "MCIN values locally diminish below 10 J kg-1". A value of 10 J kg-1 corresponds to a vertical velocity of 4.5 m s-1 that an air parcel should have to overcome the CIN barrier. It would be more meaningful to examine a lower threshold of CIN, e.e.g 1 J kg-1 corresponding to a vertical velocity of 1.4 m s-1

Answer: Following the comment, we analyzed a lower threshold of CIN reduced to 1 J kg-1. To do this, we changed the CIN figures (current Figs. 8 and 13) so that they better present the CIN values in the interval from 0 to 15 J kg-1. The analysis indicates that both non-perturbed (EX and EM, not shown in the manuscript but shown in the Supplement) and perturbed forecasts (e.g., EX0 and EM0, shown) feature CIN values in the range of 1 J kg-1 in the convection area near Mikołajki. Consequently, the relevant text of the manuscript was modified as follows:

**Section 4.1, current lines 340-342:**

"Local maxima of MCAPE reach locally 2900 and 3300 J kg-1 south of Mikołajki at 11:30 and 13:30, respectively, while MCIN immediately south-west of Mikołajki attains values below 10 J kg-1 and close to 1 J kg-1 already from 11:00 (not shown)."

**Section 4.3, current lines 431-434:**

"The MCAPE and MCIN spatial distributions in the CI area become grainy at 10:00 with local MCAPE maxima notably larger compared to the undisturbed environment (up to 2800 J kg-1 at 10:00 and 3300 J kg-1 at 11:30 in EX0). MCIN values locally diminish to about 1 J kg-1 at 10:00."

**Section 5.2, current lines 562-563:**

"Values of local MCIN minima (around 1 J kg-1) and local MCAPE maxima are similar in both ensembles."

**Reviewer 3**

This paper addresses a case study of a rapidly developing bow echo over northeastern Poland on 21 August 2007. The issue addressed is the tendency of convection-permitting models (CPMs) to that represent convection with relatively poor resolution to suppress the initiation of convection.

The case study is a good example of a major, high-impact, bow echo over Poland. This was chosen because of its high impact, but, because it occurred in 2007, it leads to a numerical experiment which is relatively unrealistic in the context of contemporary convection-permitting NWP systems. In particular, the hind-cast is spun up from ERA 5 reanalysis. This is a good choice for 2007, but the analysis is relatively low-resolution and will contain none of smaller scale variability that a realistic continuously running assimilation cycle with a CPM would have.

Answer: As discussed in our Introduction, the problems with insufficient representation of convection initiation are common for contemporary convective-scale NWP using current assimilation methods, especially for weak external forcings (Kühnlein et al. 2014; Clark et al. 2016; Hirt et al. 2019; referenced in Introduction). The issue was considered sufficiently important that it motivated a number of studies on alleviating the problem via increasing small-scale variability of the flow (Hirt et al. 2019, Puh et al. 2023, Zeng et al. 2020, Clark et al. 2021, Flack et al. 2021; referenced in the Introduction). Thus, the insufficient model-represented small-scale flow variability encountered in our study also concerns contemporary convective-scale NWP.

**Additional reference:**

Flack, D. L. A., Clark, P. A., Halliwell, C. E., Roberts, N. M., Gray, S. L., Plant, R. S., and H. W. Lean: A physically based stochastic boundary layer perturbation scheme. Part II: Perturbation growth within a superensemble framework, J. Atmos. Sci., 78, 747–761, doi:10.1175/JAS-D-19-0292.1, 2021.

This deficiency is addressed to a small extent by an assimilation cycle using surface observations; however, as one might expect, this proves only partially effective and experiments are performed essentially by adding random noise at the grid-scale to establish whether convection forms and organises.

Answer: As implied by the Reviewer, the assimilation of surface observations with a characteristic distance of tens of kilometers (70 km) could not provide information on the flow variability on smaller scales. Its sole purpose was to improve larger scale environmental conditions in the atmosphere.

As a numerical experiment, this paper is interesting and worth publishing. However, it's physical motivation is poorly justified and not well explained. They claim that they have

designed a CI scheme and "Its idea is to use near-grid-scale temperature perturbations, possibly resembling those physically developing in CBL, and allow the model to explicitly represent further upscale growth of the perturbations."

They recognise that such schemes already exist but theirs has very different properties that are not explained. In particular, their scheme is not just a stochastic perturbation scheme!

They use "temperature perturbations with a realistic amplitude of about 1.0-1.5 °C" – indeed "The amplitude  $\Delta T$  of the temperature perturbation is drawn from the Gaussian distribution with a mean of 1.25 °C and standard deviation of 0.5 °C". Thus it is biassed! They are essentially correcting a bias of 1 degree before even adding random noise. This is not just a stochastic perturbation scheme but also a bias correction scheme. Subfilter perturbations must have zero mean!

Answer: Thank you for the general opinion. Yes, the Reviewer is right that the stochastic CI scheme we applied has a positive bias, while the stochastic schemes generating small-scale temperature perturbations, like PSP, are unbiased so that they do not add energy to the atmospheric system. The stochastic method we use adds such energy, and we have acknowledged the fact in our original manuscript, stating that potential further developments of the proposed scheme should be improved for energy conservation. In this respect, our experiment follows an exploratory study of Zeng et al. 2020 using (nonstochastic) warm bubbles of 1.5 °C temperature perturbation amplitudes, with even larger horizontal sizes (10 km radius), and without compensation for the introduced temperature bias.

The net effect of heating introduced by our CI scheme, which we activate between 09:30 and 11:30, can be assessed by comparing 2-m temperatures over northeastern Poland (comparison area located between 21.0 and 23.0 degrees east and between 53.0 and 54.4 degrees north; it is a large part of the perturbation area). The spatially averaged difference between an experiment using CI perturbations (EX0) and the unperturbed one (EX) increases with time from 0.3 °C at 10:00 to 0.6 °C at 11:00, and to 0.8 °C at 12:00. These numbers are within a typical O(1 °C) error of the NWP models for this variable.

Using the unbiased scheme is partly justified as it compensates to some degree for an existing negative temperature bias of the EX forecast in the deep convection area. Of the two weather stations representative for the area, the positive effect of the CI bias was especially pronounced at Kętrzyn where at 10:00 the application of CI improves the 2m temperature bias from -1.5 in the EX experiment to -1.3 °C in the EX0 forecast, at 11:00 improves it from -0.8 to -0.2 °C, and at 12:00 improves the bias from -0.5 to 0.2 °C. The effect is more ambiguous at Mikołajki, where at 10:00 the bias is in the range of -0.4 °C for both experiments, at 11:00 the positive bias of 0.4 °C for EX is increased to 0.7 °C for EX0, and at 12:00 the bias is modified from -0.1 to 0.5 °C. Hence, the effect of the CI bias for the overall bias of the simulations, if not improving it (as it does for Kętrzyn), still keeps it limited in the range of O(1 °C), as is seen for Mikołajki.

However, we fully agree with the Reviewer that potential future applications of the proposed scheme should be unbiased, e.g., by adding a bias correction procedure. It may reflect that, physically, the temperature excess within the perturbation must be provided at the cost of

the temperature of the surrounding air. Therefore, every release of a perturbation thermal within a model could be accompanied by a (smaller in value) reduction of air temperature in the grid points of its surrounding, so that the summary temperature perturbation within the thermal and its surrounding is zero. Moreover, in analogy and consistency with the stochastic construction of the thermals, the spatial distribution of the negative temperature perturbations may be defined in a random way using a procedure that has the negative bias compensating that of the thermal.

**The following text was added to the manuscript (lines 412-421):**

"It should be noted that this perturbation technique uses only positive temperature perturbations (like the warm bubbles experiment by Zeng et al., 2020). It is therefore biased, as it additionally heats the atmosphere (the averaged effect for 2-m T is 0.2°C at 10:00 and gradually increases to 0.8°C at 12:00 for the comparison between EX0 and EX forecasts in the convective area), breaking the energy conservation principle. That was useful in our experiment as it partly compensates for the negative temperature bias of the EX-forecast for Ketrzyn in that period. The effect for Mikołajki was more ambiguous, as at 12:00 the bias of -0.1°C was modified to 0.5°C. However, potential future applications of the method should be unbiased, e.g., by introducing compensating negative temperature perturbations in the surroundings of the positive temperature perturbations. If the compensating area is sufficient, the compensating perturbations may have absolute values smaller than the positive perturbations and may also be defined in a stochastic way."

Furthermore the amplitude of their perturbations is at least an order of magnitude larger (or even two) than published schemes. This is not justified – as explained by Kober and Craig (2016) and Clark et al (2021), perturbations "physically developing in CBL" must scale on the convective temperature scale, which is typically  $O(0.1~^{\circ}C)$  – this is the typical perturbation of a thermal. When filtered to a larger scale, the amplitude will be even smaller. Clark et al (2021) show that, in deep convective situations, the continuous application of realistic CBL perturbations can lead to variability on the near grid scale that is indeed  $O(0.5~^{\circ}C)$  after 12-36 h of upscale growth, largely through moist convection.

Answer: Thank you for the comment. It is even more relevant after we have found that due to a coding error, the actual temperature perturbations we use are more pronounced than we described in the original manuscript: while the amplitude of temperature perturbation in the core grid column of a perturbation thermal ( $\Delta T$ ) is as described in the manuscript, the temperature perturbations in the 4 grid columns adjacent to the central perturbation column have the temperature perturbation equal 0.999  $\Delta T$  (which is practically  $\Delta T$ ), and not 0.09  $\Delta T$  as assumed in the original manuscript. The difference between the perturbations can be characterized by an effective horizontal size of the perturbations. It can be defined as the length of a square which has a uniform temperature perturbation  $\Delta T$  and has the summary temperature excess over its area equal to the summary temperature excess over all grid columns of the perturbation thermal. Actually, the summary temperature excess of the perturbation is practically 5  $\Delta T$  (one grid column with  $\Delta T$  excess and four columns with 0.999

 $\Delta T$  excess, each) and not 1.36  $\Delta T$  as assumed in the manuscript. Therefore, the actual effective horizontal size of the perturbation thermals is 2.2  $\Delta x$  (where  $\Delta x$  is the horizontal grid size of 2.2 km, and the metric size is 4.9 km) and not 1.2  $\Delta x$  (2.6 km) as assumed in our original manuscript. This actual size is still a near-grid-size value and is characteristic of the observed 3-to-5-km-size convective cells, which develop from 1-3-km-sized boundary layer thermals before precipitation events (Marquis et al. 2021). That near-grid size is also much less than the effective resolution size of 7  $\Delta x$ .

As for the magnitude of the applied perturbations, the Reviewer is right that typical temperature perturbations within the convective boundary layer scale like the convective temperature scale. However, the main role in deep convection initiation is likely played not by typical temperature perturbations, but by well developed thermals characterized by a larger spatial extent (Marquis et al. 2021). It is best to assess their characteristics using observational data. Williams and Hacker (1993) use aircraft measurements over land, showing that convective boundary layer thermals, having vertical and horizontal sizes comparable with the boundary layer depth, are characterized by temperature perturbation maxima exceeding the convective temperature scale by a factor of about 3.5 (Figs. 16, 17). An interesting example is shown in Fig. 13 of Williams and Hacker (1992), indicating a thermal of a horizontal size doubling the boundary layer depth and having a virtual temperature perturbation averaged over its length greater than the convective temperature scale by a factor of about 2 to 3, compared to its surroundings. The thermal has some internal structure with a hint of local sinking, a situation common for larger thermals (Williams and Hacker 1993). Such horizontal sizes of large boundary layer thermals over land are confirmed by Marquis et al. (2021), showing that their size is in the range of 1 to 3 km.

Moreover, it may be expected that the convective temperature scale within the boundary layer over northeastern Poland exceeds 0.1 °C, especially due to the large heterogeneity of terrain covered by a mixture of forests, lakes and agricultural fields. The effect may be assessed following Margairaz et al. (2020), who quantified the impact of surface thermal inhomogeneities on the convective temperature scale using the LES technique. For geostrophic wind speed of about 6 m/s, assessed for the convective area over northeastern Poland using a synoptic chart for 12:00 UTC of the day, Margairaz et al. (2020) found a convective temperature scale of 0.23 °C over homogeneous surface and of 0.4 °C over heterogeneous ones (Tab. 1). With lake surface temperatures in the range of 21 °C and 2-m air temperatures of about 27-28 °C, the surface temperature variability is comparable to that used by Margairaz et al. (2020), who applied its standard deviation at 5 °C. Their results may be, therefore, regarded as relevant for our study. Thus if, following the studies of William and Hacker, Marquis et al. (2021), and Margairaz et al. (2020), we accept the presence of boundary layer thermals with horizontal length of about 2-3 km and averaged temperature perturbations doubling or tripling the convective temperature scale of about 0.4 °C, the 1 °C range of their temperature perturbation can be regarded as realistic.

Following the above assessment, we may further estimate the realistic amplitude of the temperature perturbation for a 4.9 km convective cell developing from a 3-km-size thermal. The lower bound of that temperature amplitude may be estimated assuming that physically the cell's temperature perturbation results from a dilution of the thermal over an area of the convective cell, which is about 3 times larger than that of the thermal. That gives the cell's temperature perturbation about 3 times smaller than that of the thermal, which we use in our

experiment. If the cells' development also involves merging with neighboring thermals (the process indicated by William and Hacker, 1993; see also Stull, 1988; and Marquis et al., 2021), amplitudes of their temperature perturbation would be larger. Thus, cautiously, the temperature perturbation amplitudes we use are likely stronger by a factor of about 2-3 compared to their realistic values to allow the perturbations to effectively engage with the model dynamics. As pointed above, the temperature perturbations of similar amplitude (1.5°C) were used in the CI context for much larger perturbations, see Zeng et al. (2020) experiment using warm bubbles of about 10 km radius.

It can be additionally commented that in our experiments, the virtual temperature perturbations are represented by temperature-only perturbations. In future developments of the scheme, the perturbations can combine the variability of both temperature and humidity.

**The following text was implemented into the manuscript (lines 374-403):**

"At the few-km scales, coinciding with our near-grid scale, large boundary layer thermals with horizontal sizes of 1 to 3 km are observed over land (William and Hacker, 1992, 1993; Marquis et al., 2021), which in the process of convection initiation develop further into convective cells with horizontal sizes of 3 to 5 km (Marquis et al., 2021). Our first experiments used the temperature perturbations representing such large thermals and were applied to single grid columns (horizontal length of 2.2 km). Amplitudes of such perturbations can be estimated following William and Hacker (1992, 1993), showing that (virtual) temperature perturbations of large observed thermals in relation to their surrounding exceed the convective temperature scale, even by an averaged factor of 2-3 (Fig. 13 of William and Hacker, 1992). Also, for highly heterogeneous underlying surface and moderate geostrophic winds (characteristic for our case), the boundary layer convective temperature scale may be estimated at about 0.4°C (Margairaz et al., 2020). That gives the thermal temperature perturbation in the range of 1°C, which was applied for the model perturbations. Such perturbations, however, did not improve the forecast (not shown). However, if, besides perturbing a single grid column, the same temperature perturbation is applied also to the four neighbor grid columns, those perturbations substantially impact the CI and are used within this study. The effective horizontal size of those perturbations, taken as the length of a square of the same horizontal surface, is about 2.2 times the grid length (metrically 4.9 km), which coincides with the scale of 3-5-km cells of Marquis et al. (2021). The perturbations, therefore, may be interpreted as representing a near-grid scale flow variability related to such convective cells, and have sizes smaller than the model's effective resolution of about 7 horizontal grid spacings (Fig. 5 in Ziemiański et al., 2021). Vertically, the perturbations stretch up between the surface and 760 m AGL. The amplitude of the temperature perturbation is drawn from the Gaussian distribution with a mean of 1.25°C and a standard deviation of 0.5°C.

As for assessing the realistic amplitude of temperature perturbations of such convective cells, its lower bound may be estimated assuming that physically the perturbation results from a dilution of about 3-km-sized thermal over an area of the convective cell, which is about 3 times

larger than that of the thermal. That gives the cell's temperature perturbation about 3 times smaller than that of the thermal, which we use in our experiment. If the cells' development also involves merging with neighboring thermals (the process indicated by William and Hacker, 1993; see also Stull, 1988; and Marquis et al., 2021), amplitudes of their temperature perturbation would be larger. Thus, cautiously, our perturbation amplitudes are stronger by a factor of about 2-3 compared to their realistic values to allow the perturbations to effectively engage with the model dynamics. It may be noted that temperature perturbations of similar amplitude (1.5°C) were used in the CI context for much larger perturbations, see Zeng et al. (2020) experiment using warm bubbles of about 10 km radius."

**Additional references:**

Williams, A. G., and Hacker, J. M.: The composite shape and structure of coherent eddies in the convective boundary layers, Boundary-Layer Meteorol., 61, 213–245, doi:10.1007/BF02042933, 1992.

Williams, A. G., and Hacker, J. M.: Interactions between coherent eddies in the lower convective boundary layer, Boundary-Layer Meteorol., 64, 55–74, doi:10.1007/BF00705662, 1993.

Margairaz, F., Pardyjak, E. R., and Calaf, M.: Surface thermal heterogeneities and the atmospheric boundary layer: the relevance of dispersive fluxes, Boundary-Layer Meteorol., 77, 49–68, doi:10.1007/s10546-020-00544-7, 2020.

Stull, R. B.: An introduction to boundary layer meteorology, Kluwer Academic Publishers, Dordrecht, 1988.

Thus, the authors are modelling the variability that might exist in the initial state, but then go on to change them every 10 minutes (eddy turnover time). There is no physical justification for this. Others (some cited, many not) have used perturbations of this magnitude to study predictability, adding such noise to the initial state or at some critical time, but to refresh them so frequently is unphysical — either continuously refresh the perturbations with a physically realistic amplitude and let them grow for an appropriate period, or simply start with appropriate variability. Doing both is unphysical.

Answer: Our aim is exactly to "continuously refresh the perturbations" and "let them grow for an appropriate period", as advocated by the Reviewer, while using perturbation amplitudes possibly close to realistic values. The amplitudes, however, need to be amplified to compensate for an unphysical damping by the model. The scheme is intended to mimic a "continuous" production of the convective cells within the evolving convective boundary layer. As they are discrete entities, we model their production as a stochastic process where every perturbation is formed in a randomly chosen place and is left to grow (or dissipate) for the whole time of the simulation. Only a few such perturbations are produced at random sites during the 10-minute time interval (the eddy turnover time, as pointed out by the Reviewer) over the area of 10 by 10 grid points (above 480 square kilometers). During the next perturbation release period, new perturbations are formed in randomly defined places and they are also left free to grow, interacting with the flow, while

some of them may also interact with already existing perturbations if released in their vicinity, which also mimics physical reality. The reason for the 10-minute pause in the thermal release was exactly, as advocated by the Reviewer, not to "refresh" the perturbations too frequently, but to provide the already existing perturbations with some time for their undisturbed development. The pause diminishes the time-averaged number of perturbation releases to 2 such events per 480 square kilometers during a 10-minute interval. We did not test, however, whether this 10-minute pause, which can be regarded as unphysical in mimicking the actual process, is necessary for the success of the experiment. Because the averaged spatial density of the perturbations is relatively low, we consider it very likely that similar results would be obtained with an equivalent procedure without the pause where 2 thermals are released every 100 grid points of the convection area, during every 10-minute interval.

The following text informs that the CI scheme was not tested for alternative temporal distribution of the imposed perturbations (lines 666-668):

"As for the limitations of our experiment setup, we already discussed its temperature bias and perturbation amplitudes likely stronger (2-3 times) than realistic. We also did not optimize the method in terms of spatial and temporal density, nor the final amplitudes and shapes of the perturbations."

**Scientific significance:**

The results are of interest, though some major revision is required to put the perturbations used into context. I believe they are unphysical and also do more than randomly perturb the state. The fact that the pertubations are combined with a bias correction seems a major failing. The CI scheme could not be scientifically justified for use in a modern NWP system; the scientific question addressed is really the slightly more mundane 'a low-resolution CPM does suppresses initiation - how hard do we have to kick it to get the result we want'? The answer is in line with other studies, but tells us nothing about where that variability comes from or how, therefore, it should be represented.

Answer: Thank you. Most of the issues raised by the Reviewer (including the problem of bias correction and whether the perturbations can be considered physical) are discussed above, and the manuscript was extensively revised accordingly.

As rightly pointed by the Reviewer, our model simulations by construction (mainly because of the model's diffusive properties discussed in the next paragraph) are not able to fully represent the flow variability at near-grid scales. Thus, they are not able to "tell us where that variability comes from". Our proposition is rather to apply already existing knowledge on that variability, also based on observational studies, and use it for the development of the perturbation strategy. As for the scientific justification of our CI scheme, we argued above that the effective size of the temperature perturbations corresponds with the scale of flow variability associated with the presence of convective cells developing from large boundary layer thermals, according to Marquiraz et al. (2021). Therefore, the use of temperature perturbations of such a horizontal scale is physically justified.

As for perturbation amplitudes, we estimated above that they likely need to be stronger than physically justified (by a factor of 2-3). In this sense, the Reviewer is right that our experiment also tests how hard we need to perturb the model so that it effectively engages with such small-scale perturbations despite its numerical diffusion. The diffusion is strongest at the smallest spatial scales of the model, a behavior confirmed by the Fourier analysis of the simple diffusion equation. The analysis shows that the amplitudes of the Fourier modes of the flow are exponentially damped in time with the exponent proportional to the squared wave number of the mode. The practical effects on model results can be diagnosed by the spectral analysis of solutions' energy following Skamarock (2004). He uses the notion of "effective resolution", which is the characteristic horizontal length at which the spectrum of model-represented flow at small scales starts to diverge from the expected inclination of k\(^{\}(-5/3). The flow modes with horizontal sizes below the effective resolution size are damped by the model; the stronger, the smaller their size is. In consequence, a paradigm was accepted that the model perturbations should have horizontal sizes of at least effective resolution, and the vast experience gained with ensemble forecasting confirmed that practice (see a discussion in Palmer 2019, cited in our manuscript). Our experiment tests that paradigm for the convective moist dynamics and smaller-scale perturbations, the size of which does not represent large boundary layer thermals, as assumed in the original manuscript, but rather flow variability associated with convective cells developing from such thermals, following Marquiraz et al. (2021). As discussed above, such perturbations allow the model to develop realistic deep and organized convection if the amplitudes of the perturbations are likely overestimated to compensate for their unphysical dissipation by the model. Such a perturbation strategy (corrected for the temperature bias present in our experiments) may still be of interest for the NWP applications as it allows continuous stir of the near-grid-scale flow variability in a way that explicitly accounts for the model dissipative properties. Therefore, we consider it reasonable if future applications of such a CI procedure will deliberately use such near-grid-scale perturbations, even with overestimated temperature amplitudes.

**The following text was implemented into the manuscript in lines 99-113:**

"The above studies (except a side-experiment by Clark et al., 2021) apply perturbation horizontal sizes of O(10 km). This is the consequence of diffusive properties of NWP models, which significantly damp the flow modes having scales smaller than the model's effective resolution size (Skamarock, 2004), usually in the range of 6 to 8 grid lengths. Thus, a paradigm was accepted, also for the PSP methods (Kober et al., 2016; Clark et al., 2021), that the model perturbations should have horizontal sizes of at least the effective resolution size, and the vast experience of ensemble forecasting confirmed that practice (Palmer, 2019). Here, we want to experiment with a CI perturbation tactic that aims at stirring the flow variability at the near-grid scales, below the effective resolution size. That scale has a strong physical justification, being the scale of observed large boundary layer thermals with horizontal sizes of 1 to 3 km (William and Hacker, 1992, 1993; Marquis et al., 2021, and references therein; see also Grabowski, 2023) and initial convective cells with horizontal sizes of 3 to 5 km (Marquis et al., 2021). We demonstrate that such a scheme, used in a contemporary convective-scale NWP model, with likely overestimated perturbation amplitudes compensating for their unphysical damping, facilitates the numerical

representation of a high-impact, rapidly developing, isolated bow echo of Orlanski's (1975) meso-b-scale as the cold-pool-driven convective system with maximum gusts close to the observed ones, as long as correct large-scale environmental conditions are used."

**and in lines 666-672:**

"As for the limitations of our experiment setup, we already discussed its temperature bias and perturbation amplitudes likely stronger (2-3 times) than realistic. We also did not optimize the method in terms of spatial and temporal density, nor the final amplitudes and shapes of the perturbations. We think that the proposed CI scheme may be useful for deep convection studies. Such a perturbation strategy, corrected for its temperature bias, may still be of interest for the NWP applications because it allows continuous stirring of near-grid-scale variability in a way that accounts for the model's dissipative properties. We consider it reasonable that future applications of such a CI procedure will deliberately use near-grid-scale but overestimated temperature perturbations."

**Scientific quality:**

Most of the paper is scientifically well executed and well argued, but the basis of the CI scheme needs to be revised, claims that it is 'realistic' justified, and the role of bias correction explained.

Answer: Thank you. The manuscript was revised according to the Reviewer's comments, as discussed above.

**Presentation quality:**

On the whole the quality is acceptable, but I found navigating the maps difficult, because what landmarks there are (mainly country borders) are very hard to see,

Answer: In response to the comment, Figures 1, 2, 3, 4, 5, 6, 12 were corrected for country borders.

Minor comments

P2

39: Some more comment should be made on the deficiencies RKW. It is certainly not a complete theory.

Answer: The text of the manuscript was extended for a brief evaluation of the RKW theory (lines 39-47):

"A theory by Rotunno et al. (1988) (known as the RKW theory) explains the systems' organization and persistence via the approximate balance of horizontal vorticity of the environmental flow and of the sufficiently strong cold pool flow at the pool's leading edge.

The balance forces deep lifting and a formation of new convective cells at the leading edge, making bow echoes cold-pool-driven (Coniglio et al., 2005) systems. Further research confirmed the validity of the RKW theory for idealized systems (see Bryan et al., 2006). The studies of real events also confirm the presence of the RKW mechanism for strong low-level shears, while indicating that also the presence of a notable shear above may lead to the development of such severe and persistent systems (Stensrud et al., 2005; Weisman and Rotunno, 2005; Cognilio et al., 2012; Kirshbaum et al., 2025)."

**50: Have Bow echoes only been successfully simulated over US and Poland?**

Answer: We cited the US publications because of the vast number of bow echo studies over the US. Europe and Poland were mentioned because of the geographical proximity to our case study. The text was extended to include bow echo simulations over Africa (Diongue et al. 2002) and Asia (Meng et al. 2012, Xu et al. 2024), lines 54-58:

"Numerical models have already been used for successful numerical case studies of bow echoes developing over the US (e.g., Weisman et al., 2013; Xu et al., 2015; Parker et al., 2020; Liu et al., 2023) and Europe (Toll et al., 2015; Mathias et al., 2017), including Poland (Taszarek et al., 2019; Figurski et al., 2021; Kolonko et al., 2023; Mazur and Duniec, 2023), Africa (Diongue et al., 2002), and Asia (Meng et al., 2012; Xu et al., 2024)."

**Additional references:**

Diongue, A., Lafore, J.-P., Redelsperger, J.-L., and Roca, R.: Numerical study of a Sahelian synoptic weather system: Initiation and mature stages of convection and its interactions with the large-scale dynamics, Q. J. R. Meteorol. Soc., 128, 1899-1927, doi:10.1256/003590002320603467, 2002.

Meng, Z., Zhang, F., Markowski, P., Wu, D., and Zhao, K., 2012: A Modeling study on the development of a bowing structure and associated rear inflow within a squall line over South China, J. Atmos. Sci., 69, 1182–1207, doi:10.1175/JAS-D-11-0121.1., 2012.

Xu, X., Ju, Y., Liu, Q., Zhao, K., Xue, M., Zhang, S., Zhou, A., Wang, Y., and Tang, Y.: Dynamics of two episodes of high winds produced by an unusually long-lived quasi-linear convective system in South China, J. Atmos. Sci., 81, 1449–1473, doi:10.1175/JAS-D-23-0047.1., 2024.

64: At most 0.25 km grid spacing, not at least.

Answer: The sentence was corrected, accordingly.

84: 'the scheme' – PSP is not one scheme.

Answer: The phrase "the PSP" is used instead of "the scheme".

86 'Random representations of CI' not yet explained.

Answer: The phrase "driven only by randomly generated CI perturbations" was used instead.

103 Novelty of using ERA 5 with DA. Explain the motivation more, and discuss the scales of variability addressed by the DA.

Answer: The motivation was explained, and the scales were discussed in lines 126-132:

"An additional novelty is in augmenting the results of global ERA5 reanalysis with mesoscale data assimilation (DA), using available surface observations. That is inspired by operational procedures of augmenting global DA with regional DA for regional NWP (Gustafsson et al., 2018; Baldauf et al., 2011; Bučánek and Brožková, 2017; Müller et al., 2017). However, while regional DA mainly provides information on possibly small-scale flow variability, it is not the case here with the ERA5 31-km horizontal grid size and a 70 km characteristic distance between weather stations over Poland. Instead, we aim at a correction of larger-scale systematic temperature errors over northeastern Poland. We use operational COSMO nudging (Schraff and Hess, 2021) for that purpose."

110 Explain the meaning of 'realistic environmental conditions'.

Answer: The phrase was extended to "realistic environmental conditions (in terms of CAPE, CIN, and increased low-level vertical wind shear).".

P5

Fig 1: Hard to see maps, especially as a-c have different domains compared with c-f.

Answer: Following the comment, new Figs. 1 and 2 were prepared. Fig. 1 shows a larger domain and Fig. 2 shows a smaller domain. Additionally, following the request of Reviewer 1, Fig. 1 was reduced to a single panel showing 500 hPa surface.

P10

258: Single moment microphysics. How important is this to organisation?

Answer: The issue is outside of the scope of the manuscript. It was, however, analysed within the Damian Wójcik PhD study (Wójcik, 2021). In short, with application of a 2-moment microphysic scheme (Seiferd and Beheng, 2006) for the EM0 forecast, the bow-shaped organization of the convective system is less evident and maximum gusts are weaker (33 versus 38 m/s) compared to the results with the single moment microphysics.

Seifert, A. and K. D. Beheng, 2006: A two-moment cloud microphysics parameteri- zation for mixed-phase clouds. Part 1: Model description. *Meteorology and Atmo- spheric Physics*, **92 (1-2)**, 45–66, doi:10.1007/s00703-005-0112-4.

**P11**

No mention of cloud scheme. Probably Sommeria-Deardorff, which may slow initiation.

Answer: The Reviewer is right, COSMO uses Sommeria-Deardorf statistical cloud scheme. That information is implemented into the revised text, lines 273-277:

"C-2 and C-7 apply a single-moment cloud microphysical scheme with prognostic ice and snow (Reinhardt and Seifert, 2006) and with additional prognostic graupel for C-2, the multilevel TERRA soil model (Heise et al., 2003) and 2.5-moment turbulence scheme with an advection of turbulent kinetic energy (Mellor and Yamada, 1982; Raschendorfer, 2001), along with a variant of the statistical cloud scheme of Sommeria and Deardorf (1977; see also Doms et al., 2021)."

**P12**

280: "the realistic 280 evolutions of 2-m T and Td good indicators of realistic temperature and humidity profiles across the CBL". Some aspects, perhaps, but probably not a good indicator of cloud-base conditions. Level 2.5 MY schemes have issues with entrainment.

Answer: The text is amended (lines 294-297):

"Within the well-mixed CBL, these profiles can be approximated as functions of 2-m T and  $T_d$  (McGinley, 1986) which makes the realistic evolutions of 2-m T and  $T_d$  good indicators of realistic temperature and humidity profiles across the CBL, probably except the cloud-base conditions."

**P13**

299: How are soil temperature measurements corrected for surface altitude in the Cressman analysis?

Answer: The nudging area has relatively small variability of surface altitude (most of the stations have altitude between 100 and 200 m above msl, except Elblag with 40 m altitude

and Lublin with 238 m altitude; the latter two stations are located on opposite extremes of the nudging area), so we did not correct soil temperature measurements for surface altitudes. Following the major comment 2 of Reviewer 2, we removed the information on the technical aspects of soil temperature nudging from the manuscript.

**What about soil moisture errors?**

Answer: There are no soil moisture measurements (line 330 of the initial text, and 326-327 of the revised text) available, so the moisture errors are unknown.